# Reevaluating the neural noise in dyslexia using biomarkers from electroencephalography and high-resolution magnetic resonance spectroscopy

Agnieszka Glica[1], Katarzyna Wasilewska[1], Julia Jurkowska[2], Jarosław Żygierewicz[2], Bartosz Kossowski[3], Katarzyna Jednoróg[1]*

[1]Laboratory of Language Neurobiology, Nencki Institute of Experimental Biology, Polish Academy of Sciences, Warsaw, Poland; [2]Faculty of Physics, University of Warsaw, Warsaw, Poland; [3]Laboratory of Brain Imaging, Nencki Institute of Experimental Biology, Polish Academy of Sciences, Warsaw, Poland

**\*For correspondence:**
k.jednorog@nencki.edu.pl

**Competing interest:** The authors declare that no competing interests exist.

## eLife Assessment

This study empirically investigates the neural noise hypothesis of developmental dyslexia using electroencephalography (EEG) during a spoken language task and 7T magnetic resonance spectroscopy (MRS). The **convincing** findings indicate no evidence of an imbalance between excitatory and inhibitory (E/I) brain activity in adolescents and young adults with dyslexia compared to controls, thereby challenging the neural noise hypothesis. This research is **valuable** for advancing our understanding of the neural mechanisms underlying dyslexia and offers broader insights into the neural processes involved in reading development.

**Abstract** The neural noise hypothesis of dyslexia posits an imbalance between excitatory and inhibitory (E/I) brain activity as an underlying mechanism of reading difficulties. This study provides the first direct test of this hypothesis using both electroencephalography (EEG) power spectrum measures in 120 Polish adolescents and young adults (60 with dyslexia, 60 controls) and glutamate (Glu) and gamma-aminobutyric acid (GABA) concentrations from magnetic resonance spectroscopy (MRS) at 7T MRI scanner in half of the sample. Our results, supported by Bayesian statistics, show no evidence of E/I balance differences between groups, challenging the hypothesis that cortical hyperexcitability underlies dyslexia. These findings suggest that alternative mechanisms must be explored and highlight the need for further research into the E/I balance and its role in neurodevelopmental disorders.

## Introduction

According to the neural noise hypothesis of dyslexia, reading difficulties stem from an imbalance between excitatory and inhibitory (E/I) neural activity (*Hancock et al., 2017*). The hypothesis predicts increased cortical excitation leading to more variable and less synchronous neural firing. This instability supposedly results in disrupted sensory representations and impedes phonological awareness

and multisensory integration skills, crucial for learning to read (*Hancock et al., 2017*). Yet, studies testing this hypothesis are lacking.

The non-invasive measurement of the E/I balance can be derived through assessment of glutamate (Glu) and gamma-aminobutyric acid (GABA) neurotransmitters concentration via magnetic resonance spectroscopy (MRS) (*Finkelman et al., 2022*) or through various E/I estimations from the electroencephalography (EEG) signal (*Ahmad et al., 2022*).

MRS measurements of Glu and GABA yielded conflicting findings. Higher Glu concentrations in the midline occipital cortex correlated with poorer reading performance in children (*Del Tufo et al., 2018*; *Pugh et al., 2014*), while elevated Glu levels in the anterior cingulate cortex (ACC) corresponded to greater phonological skills (*Lebel et al., 2016*). Elevated GABA in the left inferior frontal gyrus was linked to reduced verbal fluency in adults (*Nakai and Okanoya, 2016*), and increased GABA in the midline occipital cortex in children was associated with slower reaction times in a linguistic task (*Del Tufo et al., 2018*). However, notable null findings exist regarding dyslexia status and Glu levels in the ACC among children (*Horowitz-Kraus et al., 2018*) as well as Glu and GABA levels in the visual and temporo-parietal cortices in both children and adults (*Kossowski et al., 2019*).

Both beta (~13–28 Hz) and gamma (>30 Hz) oscillations may serve as E/I balance indicators (*Ahmad et al., 2022*), as greater GABA-ergic activity has been associated with greater beta power (*Jensen et al., 2005*; *Porjesz et al., 2002*) and gamma power or peak frequency (*Brunel and Wang, 2003*; *Chen et al., 2017*). Resting-state analyses often reported non-significant beta power associations with dyslexia (*Babiloni et al., 2012*; *Fraga González et al., 2018*; *Xue et al., 2020*), however, one study indicated lower beta power in dyslexic compared to control boys (*Fein et al., 1986*). Mixed results were also observed during tasks. One study found decreased beta power in the dyslexic group (*Spironelli et al., 2008*), while the other noted increased beta power relative to the control group (*Rippon and Brunswick, 2000*). Insignificant relationship between resting gamma power and dyslexia was reported (*Babiloni et al., 2012*; *Lasnick et al., 2023*). When analyzing auditory steady-state responses, the dyslexic group had a lower gamma peak frequency, while no significant differences in gamma power were observed (*Rufener and Zaehle, 2021*). Essentially, the majority of studies in dyslexia examining gamma frequencies evaluated cortical entrainment to auditory stimuli (*Lehongre et al., 2011*; *Marchesotti et al., 2020*; *Van Hirtum et al., 2019*). Therefore, the results from these tasks do not provide direct evidence of differences in either gamma power or peak frequency between the dyslexic and control groups.

The EEG signal comprises both oscillatory, periodic activity, and aperiodic activity, characterized by a gradual decrease in power as frequencies rise (1/*f* signal) (*Donoghue et al., 2020*). Recently recognized as a biomarker of E/I balance, a lower exponent of signal decay (flatter slope) indicates a greater dominance of excitation over inhibition in the brain, as shown by the simulation models of local field potentials, ratio of AMPA/GABA$_a$ synapses in the rat hippocampus (*Gao et al., 2017*), and recordings under propofol or ketamine in macaques and humans (*Gao et al., 2017*; *Waschke et al., 2021*). However, there are also pharmacological studies providing mixed results (*Colombo et al., 2019*; *Salvatore et al., 2024*). Nonetheless, the 1/*f* signal has shown associations with various conditions putatively characterized by changes in E/I balance, such as early development in infancy (*Schaworonkow and Voytek, 2021*), healthy aging (*Voytek et al., 2015*), and neurodevelopmental disorders like ADHD (*Ostlund et al., 2021*), autism spectrum disorder (*Manyukhina et al., 2022*), or schizophrenia (*Molina et al., 2020*). Despite its potential relevance, the evaluation of the 1/*f* signal in dyslexia remains limited to one study, revealing flatter slopes among dyslexic compared to control participants at rest (*Turri et al., 2023*), thereby lending support to the notion of neural noise in dyslexia.

Here, we examined both EEG (1/*f* signal, beta, and gamma oscillations during both rest and a spoken language task) and MRS (Glu and GABA) biomarkers of E/I balance in participants with dyslexia and age-matched controls. The neural noise hypothesis predicts flatter slopes of 1/*f* signal, decreased beta and gamma power, and higher Glu concentrations in the dyslexic group. Furthermore, we tested the relationships between different E/I measures. Flatter slopes of 1/*f* signal should be related to higher Glu level, while enhanced beta and gamma power to increased GABA level.

# Results

## No evidence for group differences in the EEG E/I biomarkers

We recruited 120 Polish adolescents and young adults – 60 with dyslexia diagnosis and 60 controls matched in sex, age, and family socio-economic status. The dyslexic group scored lower in all reading and reading-related tasks and higher in the Polish version of the Adult Reading History Questionnaire (ARHQ-PL) (**Bogdanowicz et al., 2015**), where a higher score indicates a higher risk of dyslexia (see **Table 1**). Although all participants were within the intellectual norm, the dyslexic group scored lower on the IQ scale (including nonverbal subscale only) than the control group. However, the Bayesian statistics did not provide evidence for the difference between groups in the nonverbal IQ.

The EEG signal was recorded at rest and during a spoken language task, where participants listened to a sentence and had to indicate its veracity. In the initial step of the analysis, we analyzed the aperiodic (exponent and offset) components of the EEG spectrum. The exponent reflects the steepness of the EEG power spectrum, with a higher exponent indicating a steeper signal; while the offset represents a uniform shift in power across frequencies, with a higher offset indicating greater power across the entire EEG spectrum (**Donoghue et al., 2020**).

Due to a technical error, the signal from one person (a female from the dyslexic group) was not recorded during most of the language task and was excluded from the analyses. Hence, the results are provided for 119 participants – 59 in the dyslexic and 60 in the control group.

First, aperiodic parameter values were averaged across all electrodes and compared between groups (dyslexic, control) and conditions (resting state, language task) using a 2×2 repeated measures ANOVA. Age negatively correlated both with the exponent ($r=-0.27$, $p=0.003$, $BF_{10}=7.96$) and offset ($r=-0.40$, $p<0.001$, $BF_{10}=3174.29$) in line with previous investigations (**Cellier et al., 2021**; **McSweeney et al., 2021**; **Schaworonkow and Voytek, 2021**; **Voytek et al., 2015**), therefore we included age as a covariate. Post hoc tests are reported with Bonferroni corrected p-values.

For the mean exponent, we found a significant effect of age ($F(1,116) = 8.90$, $p=0.003$, $\eta^2_p = 0.071$, $BF_{incl} = 10.47$), while the effects of condition ($F(1,116) = 2.32$, $p=0.131$, $\eta^2_p = 0.020$, $BF_{incl} = 0.39$) and group ($F(1,116) = 0.08$, $p=0.779$, $\eta^2_p = 0.001$, $BF_{incl} = 0.40$) were not significant and Bayes factor did not provide evidence for either inclusion or exclusion. Interaction between group and condition ($F(1,116) = 0.16$, $p=0.689$, $\eta^2_p = 0.001$, $BF_{incl} = 0.21$) was not significant and Bayes factor indicated against including it in the model.

For the mean offset, we found significant effects of age ($F(1,116) = 22.57$, $p<0.001$, $\eta^2_p = 0.163$, $BF_{incl} = 1762.19$) and condition ($F(1,116) = 23.04$, $p<0.001$, $\eta^2_p = 0.166$, $BF_{incl} >10,000$) with post hoc comparison indicating that the offset was lower in the resting-state condition ($M=-10.80$, $SD = 0.21$) than in the language task ($M=-10.67$, $SD = 0.26$, $p_{corr}<0.001$). The effect of group ($F(1,116) = 0.00$, $p=0.964$, $\eta^2_p = 0.000$, $BF_{incl} = 0.54$) was not significant while Bayes factor did not provide evidence for either inclusion or exclusion. Interaction between group and condition was not significant ($F(1,116) = 0.07$, $p=0.795$, $\eta^2_p = 0.001$, $BF_{incl} = 0.22$) and Bayes factor indicated against including it in the model.

Next, we restricted analyses to language regions and averaged exponent and offset values from the frontal electrodes corresponding to the left (F7, FT7, FC5) and right inferior frontal gyrus (F8, FT8, FC6), as well as temporal electrodes, corresponding to the left (T7, TP7, TP9) and right superior temporal sulcus, STS (T8, TP8, TP10) (**Giacometti et al., 2014**; **Scrivener and Reader, 2022**). A 2×2×2×2 (group, condition, hemisphere, region) repeated measures ANOVA with age as a covariate was applied. Power spectra from the left STS at rest and during the language task are presented in **Figure 1A and C**, while the results for the exponent, offset, and beta power are presented in **Figure 1B and D**.

For the exponent, there were significant effects of age ($F(1,116) = 14.00$, $p<0.001$, $\eta^2_p = 0.108$, $BF_{incl} = 11.46$) and condition ($F(1,116) = 4.06$, $p=0.046$, $\eta^2_p = 0.034$, $BF_{incl} = 1.88$), however, Bayesian statistics did not provide evidence for either including or excluding the condition factor. Furthermore, post hoc comparisons did not reveal significant differences between the exponent at rest ($M=1.51$, $SD = 0.17$) and during the language task ($M=1.51$, $SD = 0.18$, $p_{corr} = 0.546$). There was also a significant interaction between region and group, although Bayes factor indicated against including it in the model ($F(1,116) = 4.44$, $p=0.037$, $\eta^2_p = 0.037$, $BF_{incl} = 0.25$). Post hoc comparisons indicated that the exponent was higher in the frontal than in the temporal region both in the dyslexic ($M_{frontal} = 1.54$, $SD_{frontal} = 0.15$, $M_{temporal} = 1.49$, $SD_{temporal} = 0.18$, $p_{corr}<0.001$) and in the control group ($M_{frontal} = 1.54$, $SD_{frontal} = 0.17$, $M_{temporal} = 1.46$, $SD_{temporal} = 0.20$, $p_{corr}<0.001$). The difference between groups was not

**Table 1.** Demographic and behavioral characteristics of the entire sample of 120 participants.

| | DYS 28 F, 32 M | | CON 28 F, 32 M | | t | p | Cohen's d | BF₁₀ |
|---|---|---|---|---|---|---|---|---|
| | M | SD | M | SD | (df) | | | |
| *Demographics* | | | | | | | | |
| Age | 19.41 | 3.18 | 19.54 | 2.96 | 0.25 (118) | 0.806 | 0.05 | 0.20 |
| Mother's education (years) | 17.20 | 3.36 | 16.58 | 2.28 | −1.19 (103.89) | 0.235 | −0.22 | 0.37 |
| Father's education (years) | 16.12[a] | 3.10[a] | 17.13[a] | 3.27[a] | 1.71 (114) | 0.091 | 0.32 | 0.73 |
| IQ | 103.56[b] | 11.83[b] | 111.12 | 10.43 | 3.70 (117) | **<0.001** | 0.68 | 75.31 |
| Nonverbal IQ (scaled score) | 10.40 | 2.94 | 11.62 | 2.57 | 2.42 (118) | **0.017** | 0.44 | 2.63 |
| ARHQ-PL | 51.50 | 9.70 | 25.47 | 8.00 | −16.04 (113.87) | **<0.001** | −2.93 | >10,000 |
| *Reading and reading-related tasks* | | | | | | | | |
| Words/min | 108.38 | 20.93 | 134.57 | 13.29 | 8.18 (99.90) | **<0.001*** | 1.49 | >10,000 |
| Pseudowords/min | 56.75 | 14.16 | 83.43 | 17.04 | 9.33 (118) | **<0.001*** | 1.70 | >10,000 |
| RAN objects (s) | 32.12 | 5.11 | 28.70 | 4.43 | −3.92 (118) | **<0.001*** | −0.72 | 149.93 |
| RAN colors (s) | 35.83 | 6.82 | 31.18 | 5.73 | −4.04 (118) | **<0.001*** | −0.74 | 229.96 |
| RAN digits(s) | 19.32 | 4.61 | 16.25 | 2.94 | −4.34 (100.28) | **<0.001*** | −0.79 | 642.86 |
| RAN letters (s) | 22.70 | 4.53 | 19.68 | 3.16 | −4.23 (105.42) | **<0.001*** | −0.77 | 433.23 |
| Reading comprehension (s) | 64.47 | 20.13 | 43.72 | 9.63 | −7.20 (84.66) | **<0.001*** | −1.32 | >10,000 |
| Phoneme deletion (% correct) | 76.41 | 24.68 | 91.47 | 9.07 | 4.44 (74.66) | **<0.001*** | 0.81 | 898.25 |
| Spoonerisms phonemes (% correct) | 54.29 | 35.42 | 82.74 | 22.06 | 5.28 (98.78) | **<0.001*** | 0.96 | >10,000 |
| Spoonerisms syllables (% correct) | 46.94 | 30.61 | 73.06 | 23.98 | 5.20 (111.62) | **<0.001*** | 0.95 | >10,000 |
| Orthographic awareness (accuracy/time) | 0.33 | 0.13 | 0.53 | 0.14 | 8.12 (118) | **<0.001*** | 1.48 | >10,000 |
| Perception speed (sten score) | 3.32 | 2.04 | 4.50 | 1.67 | 3.48 (118) | **<0.001*** | 0.64 | 38.71 |
| Digits forward | 5.53 | 1.64 | 6.98 | 1.95 | 4.40 (118) | **<0.001*** | 0.80 | 792.55 |
| Digits backward | 5.25 | 1.49 | 7.33 | 2.25 | 5.99 (102.59) | **<0.001*** | 1.09 | >10,000 |

Note: DYS – dyslexic group; CON – control group; F – females, M – males. BF₁₀ – Bayes factor indicating ratio of the likelihood of an alternative hypothesis (H1) to a null hypothesis (H0). ARHQ-PL – Polish version of the Adult Reading History Questionnaire. RAN – rapid automatized naming. Boldface indicates statistical significance at p<0.05 level (uncorrected).

*Significance after Bonferroni correction for 14 planned comparisons for reading and reading-related tasks

[a]n = 58 (two participants did not provide information about the father's education)

[b]n = 59 (one participant did not attempt a verbal subtest of the scale, thus we were not able to calculate overall IQ)

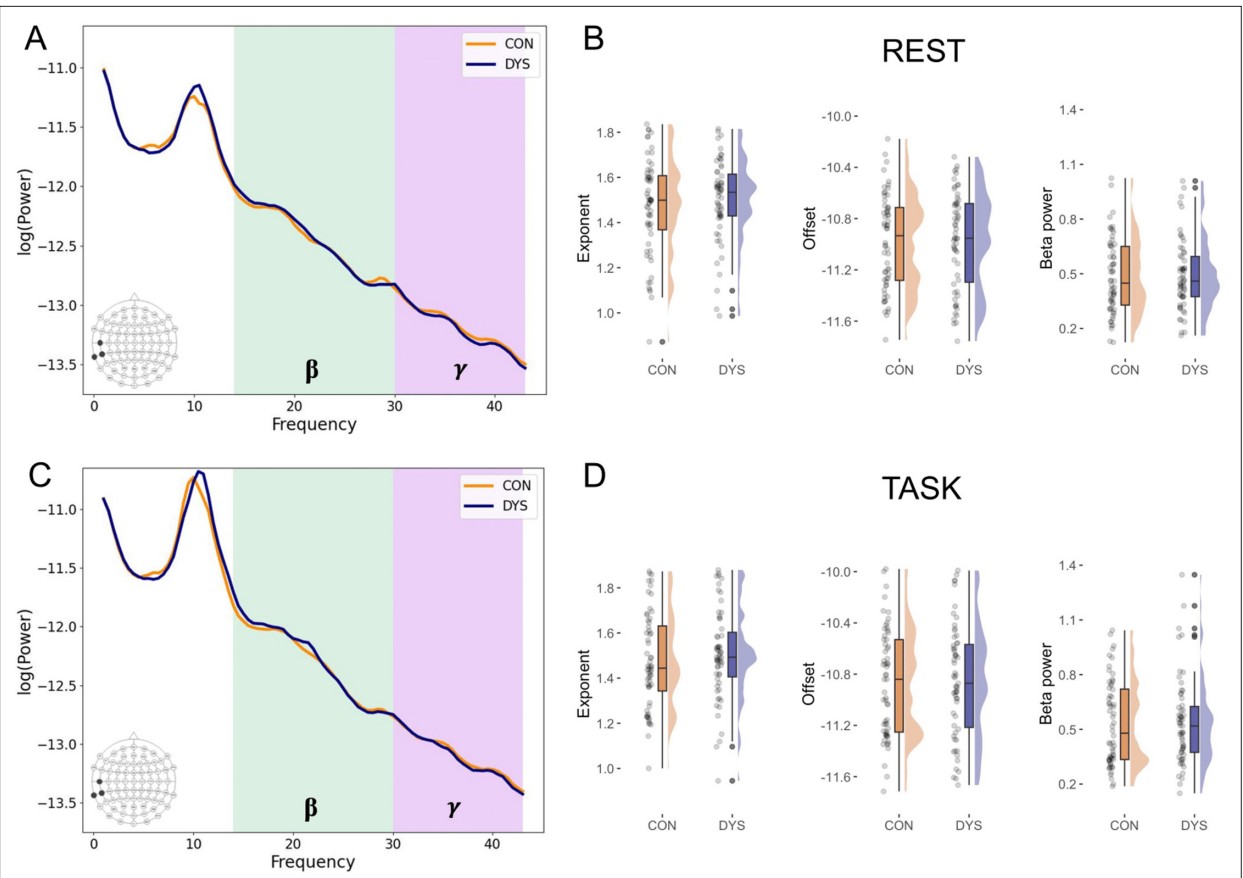

**Figure 1.** Selected results for the electroencephalography (EEG) excitatory and inhibitory (E/I) balance biomarkers. $n$ = 119 (DYS $n$ = 59, CON $n$ = 60). (**A**) Power spectral densities averaged across three electrodes (T7, TP7, TP9)corresponding to the left superior temporal sulcus (STS) separately for dyslexic (DYS) and control (CON) groups at rest and (**C**) during the language task. (**B**) Plots illustrating results for the exponent, offset, and the beta power from the left STS electrodes at rest and (**D**) during the language task.

significant either in the frontal ($p_{corr}$=0.858) or temporal region ($p_{corr}$=0.441). The effects of region ($F(1,116)$ = 1.17, p=0.282, $\eta^2_p$ = 0.010, $BF_{incl}$>10,000) and hemisphere ($F(1,116)$ = 1.17, p=0.282, $\eta^2_p$ = 0.010, $BF_{incl}$ = 12.48) were not significant, although Bayesian statistics indicated in favor of including them in the model. Furthermore, the interactions between condition and group ($F(1,116)$ = 0.18, p=0.673, $\eta^2_p$ = 0.002, $BF_{incl}$ = 3.70), and between region, hemisphere, and condition ($F(1,116)$ = 0.11, p=0.747, $\eta^2_p$ = 0.001, $BF_{incl}$ = 7.83) were not significant, however Bayesian statistics indicated in favor of including these interactions in the model. The effect of group ($F(1,116)$ = 0.12, p=0.733, $\eta^2_p$ = 0.001, $BF_{incl}$ = 1.19) was not significant, while Bayesian statistics did not provide evidence for either inclusion or exclusion. Any other interactions were not significant and Bayes factor indicated against including them in the model. Since Bayes factor suggested the inclusion of the condition*group interaction in the model, we further conducted Bayesian $t$-tests to determine whether this was driven by differences between control and dyslexic groups in either condition. The results, however, supported the null hypothesis in both the resting-state condition ($M_{DYS}$ = 1.51, $SD_{DYS}$ = 0.16, $M_{CON}$ = 1.50, $SD_{CON}$ = 0.19, $BF_{10}$=0.22) and during the language task ($M_{DYS}$ = 1.52, $SD_{DYS}$ = 0.17, $M_{CON}$ = 1.51, $SD_{CON}$ = 0.19, $BF_{10}$=0.20).

In the case of offset, there were significant effects of condition ($F(1,116)$ = 20.88, p<0.001, $\eta^2_p$ = 0.153, $BF_{incl}$>10,000) and region ($F(1,116)$ = 6.18, p=0.014, $\eta^2_p$ = 0.051, $BF_{incl}$>10,000). For the main effect of condition, post hoc comparison indicated that the offset was lower in the resting-state condition ($M$=−10.88, $SD$ = 0.33) than in the language task ($M$=−10.76, $SD$ = 0.38, $p_{corr}$<0.001), while for the main effect of region, post hoc comparison indicated that the offset was lower in the temporal ($M$=−10.94, $SD$ = 0.37) as compared to the frontal region ($M$=−10.69, $SD$ = 0.34, $p_{corr}$<0.001). There was also a significant effect of age ($F(1,116)$ = 20.84, p<0.001, $\eta^2_p$ = 0.152, $BF_{incl}$ = 0.23) and interaction

between condition and hemisphere, ($F$(1,116) = 4.35, p=0.039, $\eta^2_p$ = 0.036, $BF_{incl}$ = 0.21), although Bayes factor indicated against including these factors in the model. Post hoc comparisons for the condition*hemisphere interaction indicated that the offset was lower in the resting-state condition than in the language task both in the left ($M_{rest}$ = −10.85, $SD_{rest}$ = 0.34, $M_{task}$ = −10.73, $SD_{task}$ = 0.40, $p_{corr}$<0.001) and in the right hemisphere ($M_{rest}$ = −10.91, $SD_{rest}$ = 0.31, $M_{task}$ = −10.79, $SD_{task}$ = 0.37, $p_{corr}$<0.001) and that the offset was lower in the right as compared to the left hemisphere both at rest ($p_{corr}$<0.001) and during the language task ($p_{corr}$<0.001). The interactions between region and condition ($F$(1,116) = 1.76, p=0.187, $\eta^2_p$ = 0.015, $BF_{incl}$>10,000), hemisphere and group ($F$(1,116) = 1.58, p=0.211, $\eta^2_p$ = 0.013, $BF_{incl}$ = 1595.18), region and group ($F$(1,116) = 0.27, p=0.605, $\eta^2_p$ = 0.002, $BF_{incl}$ = 9.32), as well as between region, condition, and group ($F$(1,116) = 0.21, p=0.651, $\eta^2_p$ = 0.002, $BF_{incl}$ = 2867.18) were not significant, although Bayesian statistics indicated in favor of including them in the model. The effect of group ($F$(1,116) = 0.18, p=0.673, $\eta^2_p$ = 0.002, $BF_{incl}$<0.00001) was not significant and Bayesian statistics indicated against including it in the model. Any other interactions were not significant and Bayesian statistics indicated against including them in the model or did not provide evidence for either inclusion or exclusion. Since Bayes factor suggested the inclusion of hemisphere*group, region*group, and region*condition*group interactions in the model, we further conducted Bayesian $t$-tests to determine whether this was driven by differences between control and dyslexic groups. The results indicated in favor of the null hypothesis both in the left ($M_{DYS}$ = −10.78, $SD_{DYS}$ = 0.38, $M_{CON}$ = −10.80, $SD_{CON}$ = 0.36, $BF_{10}$=0.20) and right hemisphere ($M_{DYS}$ = −10.83, $SD_{DYS}$ = 0.32, $M_{CON}$ = −10.87, $SD_{CON}$ = 0.36, $BF_{10}$=0.24) as well as in the frontal ($M_{DYS}$ = −10.68, $SD_{DYS}$ = 0.34, $M_{CON}$ = −10.71, $SD_{CON}$ = 0.34, $BF_{10}$=0.21) and temporal regions ($M_{DYS}$ = −10.92, $SD_{DYS}$ = 0.36, $M_{CON}$ = −10.96, $SD_{CON}$ = 0.38, $BF_{10}$=0.22). Similarly, results for the region*condition*group interaction, indicated in favor of null hypothesis both in frontal ($M_{DYS}$ = −10.75, $SD_{DYS}$ = 0.31, $M_{CON}$ = −10.77, $SD_{CON}$ = 0.32, $BF_{10}$=0.21) and temporal electrodes at rest ($M_{DYS}$ = −10.98, $SD_{DYS}$ = 0.34, $M_{CON}$ = −11.01, $SD_{CON}$ = 0.36, $BF_{10}$=0.22) as well as in frontal ($M_{DYS}$ = −10.61, $SD_{DYS}$ = 0.37, $M_{CON}$ = −10.64, $SD_{CON}$ = 0.37, $BF_{10}$=0.21) and temporal electrodes during the language task ($M_{DYS}$ = −10.87, $SD_{DYS}$ = 0.39, $M_{CON}$ = −10.90, $SD_{CON}$ = 0.41, $BF_{10}$=0.22).

Then, we analyzed the aperiodic-adjusted brain oscillations. Since the algorithm did not find the gamma peak (30–43 Hz) above the aperiodic component in the majority of participants, we report the results only for the beta (14–30 Hz) power. We performed a similar regional analysis as for the exponent and offset with a 2×2×2×2 (group, condition, hemisphere, region) repeated measures ANOVA. However, we did not include age as a covariate, as it did not correlate with any of the periodic measures. The sample size was 117 (DYS $n$=57, CON $n$=60) since in two participants the algorithm did not find the beta peak above the aperiodic component in the left frontal electrodes during the task.

The analysis revealed a significant effect of condition ($F$(1,115) = 8.58, p=0.004, $\eta^2_p$ = 0.069, $BF_{incl}$ = 5.82) with post hoc comparison indicating that the beta power was greater during the language task ($M$=0.53, $SD$ = 0.22) than at rest ($M$=0.50, $SD$ = 0.19, $p_{corr}$ = 0.004). There were also significant effects of region ($F$(1,115) = 10.98, p=0.001, $\eta^2_p$ = 0.087, $BF_{incl}$ = 23.71), and hemisphere ($F$(1,115) = 12.08, p<0.001, $\eta^2_p$ = 0.095, $BF_{incl}$ = 23.91). For the main effect of region, post hoc comparisons indicated that the beta power was greater in the temporal ($M$=0.52, $SD$ = 0.21) as compared to the frontal region ($M$=0.50, $SD$ = 0.19, $p_{corr}$ = 0.001), while for the main effect of hemisphere, post hoc comparisons indicated that the beta power was greater in the right ($M$=0.52, $SD$ = 0.20) than in the left hemisphere ($M$=0.51, $SD$ = 0.20, $p_{corr}$<0.001). There was a significant interaction between condition and region ($F$(1,115) = 12.68, p<0.001, $\eta^2_p$ = 0.099, $BF_{incl}$ = 55.26) with greater beta power during the language task as compared to rest significant in the temporal ($M_{rest}$ = 0.50, $SD_{rest}$ = 0.20, $M_{task}$ = 0.55, $SD_{task}$ = 0.24, $p_{corr}$<0.001), while not in the frontal region ($M_{rest}$ = 0.49, $SD_{rest}$ = 0.18, $M_{task}$ = 0.51, $SD_{task}$ = 0.22, $p_{corr}$ = 0.077). Also, greater beta power in the temporal as compared to the frontal region was significant during the language task ($p_{corr}$<0.001), while not at rest ($p_{corr}$=0.283). The effect of group ($F$(1,115) = 0.05, p=0.817, $\eta^2_p$ = 0.000, $BF_{incl}$<0.00001) was not significant and Bayes factor indicated against including it in the model. Any other interactions were not significant and Bayesian statistics indicated against including them in the model or did not provide evidence for either inclusion or exclusion. Descriptive statistics for EEG results separately for dyslexic and control groups are provided in *Appendix 1—table 1*.

Additionally, building upon previous findings which demonstrated differences in dyslexia in aperiodic and periodic components within the parieto-occipital region (*Turri et al., 2023*), we have included

analyses for the same cluster of electrodes in Appendix 1. However, in this region, we also did not find evidence for group differences either in the exponent, offset, or beta power.

## No evidence for group differences in Glu and GABA+ concentrations in the left STS

The MRS voxel was placed in the left STS, in a region showing highest activation for both visual and auditory words (compared to control stimuli) localized individually in each participant, based on an fMRI task (see *Figure 2—figure supplement 1* and *Appendix 1—table 5*, *Appendix 1—table 6*, and *Appendix 1—table 7* for the group-level results from the fMRI task). Sample voxel localization and sample MRS spectrum are presented in *Figure 2A and B*, while overlap of the MRS voxel placement across participants and MRS spectra separately for the dyslexic and control groups are presented in *Figure 2C and D*. We decided to analyze the neurometabolites' levels derived from the left STS, as this region is consistently related to functional and structural differences in dyslexia across languages (*Yan et al., 2021*). Moreover, the neural noise hypothesis of dyslexia identifies perisylvian areas as being affected by increased glutamatergic signaling, and directly predicts associations between Glu and GABA levels in the superior temporal regions and phonological skills (*Hancock et al., 2017*).

Due to financial and logistical constraints, 59 out of the 120 recruited subjects, selected progressively as the study unfolded, were examined with MRS. Subjects were matched by age and sex between the dyslexic and control groups. Due to technical issues and to prevent delays and discomfort for the participants, we collected 54 complete sessions. Additionally, four datasets were excluded based on our quality control criteria (linewidth>20 Hz and visual inspection), and three GABA+ estimates exceeded the selected CRLB threshold (>20%). Ultimately, we report 50 estimates for Glu (21 participants with dyslexia) and 47 for GABA+ and Glu/GABA+ ratios (20 participants with dyslexia). Demographic and behavioral characteristics for the subsample of 47 participants are provided in *Appendix 1—table 2*. For comparability with previous studies in dyslexia (*Del Tufo et al., 2018*; *Pugh et al., 2014*) we report Glu and GABA as a ratio to total creatine (tCr).

For each metabolite, we performed a separate univariate ANCOVA with the effect of group being tested and voxel's gray matter volume (GMV) as a covariate (see *Figure 2E*). For the Glu analysis, we also included age as a covariate, due to negative correlation between variables ($r=-0.35$, p=0.014, $BF_{10}=3.41$). The analysis revealed significant effect of GMV ($F(1,46) = 8.18$, p=0.006, $\eta^2_p = 0.151$, $BF_{incl} = 12.54$), while the effects of age ($F(1,46) = 3.01$, p=0.090, $\eta^2_p = 0.061$, $BF_{incl} = 1.15$) and group $F(1,46) = 1.94$, p=0.170, $\eta^2_p = 0.040$, $BF_{incl} = 0.63$ were not significant and Bayes factor did not provide evidence for either inclusion or exclusion.

Conversely, GABA+ did not correlate with age ($r=-0.11$, p=0.481, $BF_{10}=0.23$), thus age was not included as a covariate. The analysis revealed a significant effect of GMV ($F(1,44) = 4.39$, p=0.042, $\eta^2_p = 0.091$, $BF_{incl} = 1.64$), however Bayes factor did not provide evidence for either inclusion or exclusion. The effect of group was not significant ($F(1,44) = 0.49$, p=0.490, $\eta^2_p = 0.011$, $BF_{incl} = 0.35$) although Bayesian statistics did not provide evidence for either inclusion or exclusion.

Also, Glu/GABA+ ratio did not correlate with age ($r=-0.05$, p=0.744, $BF_{10}=0.19$), therefore age was not included as a covariate. The results indicated that the effect of GMV was not significant ($F(1,44) = 0.95$, p=0.335, $\eta^2_p = 0.021$, $BF_{incl} = 0.43$) while Bayes factor did not provide evidence for either inclusion or exclusion. The effect of group was not significant ($F(1,44) = 0.01$, p=0.933, $\eta^2_p = 0.000$, $BF_{incl} = 0.29$) and Bayes factor indicated against including it in the model.

Following a recent study examining developmental changes in both EEG and MRS E/I biomarkers (*McKeon et al., 2024*), we calculated an additional measure of Glu/GABA+ imbalance, computed as the absolute residual value from the linear regression of Glu predicted by GABA+ with greater values indicating greater Glu/GABA+ imbalance. Alike the previous work (*McKeon et al., 2024*), we took the square root of this value to ensure a normal distribution of the data. This measure did not correlate with age ($r=-0.05$, p=0.719, $BF_{10}=0.19$); thus, age was not included as a covariate. The results indicated that the effect of GMV was not significant ($F(1,44) = 0.63$, p=0.430, $\eta^2_p = 0.014$, $BF_{incl} = 0.37$) while Bayes factor did not provide evidence for either inclusion or exclusion. The effect of group was not significant ($F(1,44) = 0.74$, p=0.396, $\eta^2_p = 0.016$, $BF_{incl} = 0.39$) although Bayesian statistics did not provide evidence for either inclusion or exclusion. Descriptive statistics for MRS results separately for dyslexic and control groups are provided in *Appendix 1—table 1*.

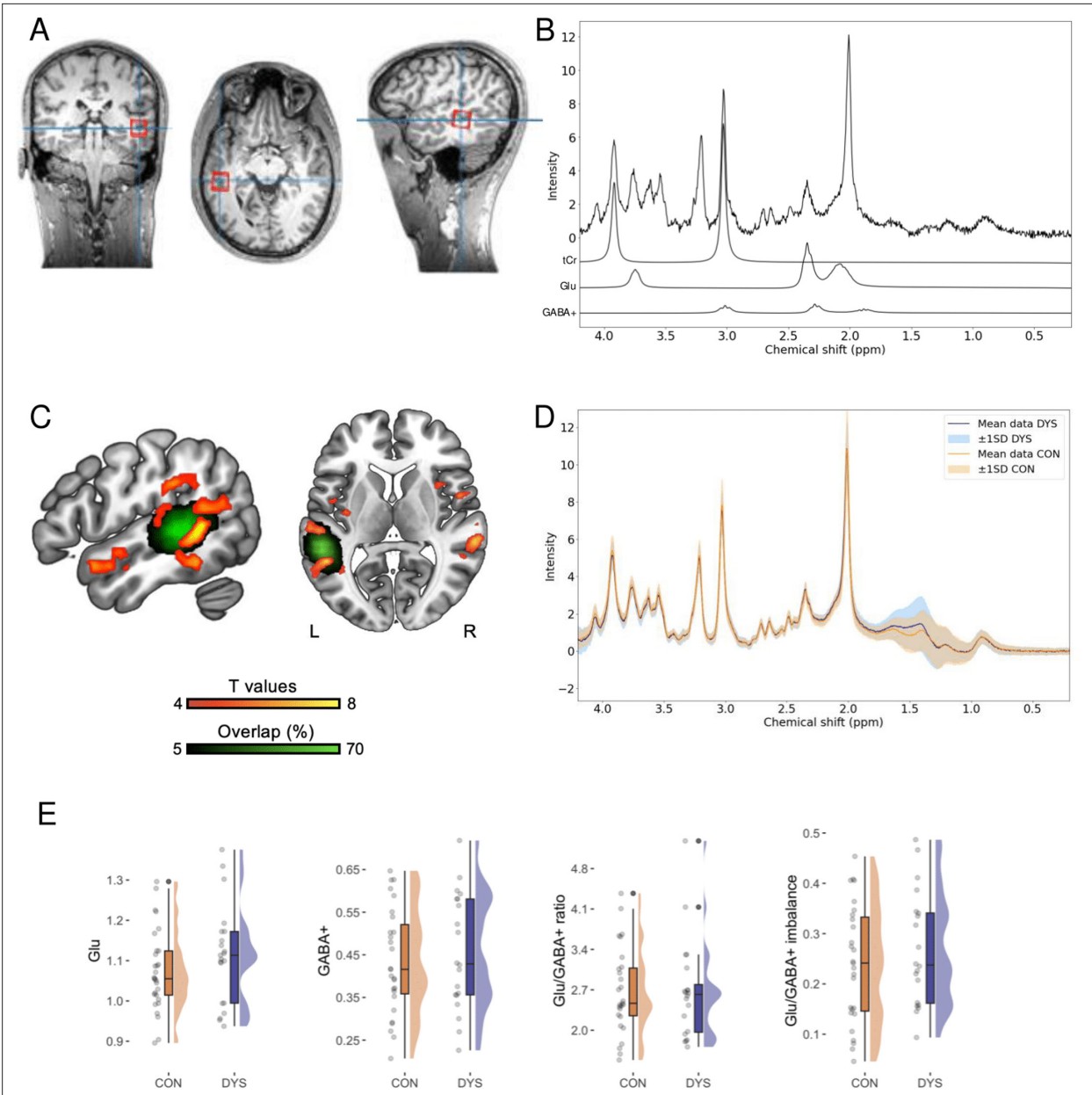

**Figure 2.** Selected results for the magnetic resonance spectroscopy (MRS) excitatory and inhibitory (E/I) balance biomarkers. (**A**) Sample MRS voxel localization. (**B**) Sample MRS spectrum with individual total creatine (tCr), Glu, and GABA+ contributions plotted below. (**C**) Group results (CON>DYS) from the fMRI localizer task for words compared to the control stimuli (p<0.05 FWE cluster threshold) and overlap of the MRS voxel placement across participants. n = 50 (DYS n = 21, CON n = 29). (**D**) MRS spectra separately for the dyslexic (DYS) and control (CON) groups. n = 50 (DYS n = 21, CON n = 29). (**E**) Plots illustrating results for the Glu, n = 50 (DYS n = 21, CON n = 29), GABA+, n = 47 (DYS n = 20, CON n = 27), Glu/GABA+ ratio, n = 47 (DYS n = 20, CON n = 27), and the Glu/GABA+ imbalance, n = 47 (DYS n = 20, CON n = 27).

The online version of this article includes the following figure supplement(s) for figure 2:

**Figure supplement 1.** Main effect of group CON>DYS for both visual (words > false fonts) and auditory runs (words > words backward) from the fMRI localizer task.

## Correspondence between Glu and GABA+ concentrations and EEG E/I biomarkers is limited

Next, we investigated correlations between Glu and GABA+ concentrations in the left STS and EEG biomarkers of E/I balance. Semi-partial correlations were performed (*Table 2*) to control for confounding variables – for Glu the effects of age and GMV were regressed, for GABA+, Glu/GABA+

**Table 2.** Semi-partial correlations between magnetic resonance spectroscopy (MRS) and electroencephalography (EEG) biomarkers of excitatory-inhibitory balance.

For Glu the effects of age and gray matter volume (GMV) were regressed, for GABA+, Glu/GABA+ ratio, and Glu/GABA+ imbalance the effect of GMV was regressed, while for exponents and offsets the effect of age was regressed.

| Variable | 1. $r$ ($BF_{10}$) | 2. | 3. | 4. | 5. | 6. | 7. | 8. |
|---|---|---|---|---|---|---|---|---|
| *EEG resting state* | | | | | | | | |
| 1. Glu | – | | | | | | | |
| 2. GABA+ | 0.32*[a] (1.82) | – | | | | | | |
| 3. Glu/GABA+ ratio | –0.08[a] (0.21) | –0.91***[a] (>10,000) | – | | | | | |
| 4. Glu/GABA+ imbalance | 0.12[a] (0.25) | 0.31*[a] (1.63) | –0.18[a] (0.37) | – | | | | |
| 5. Exponent mean (rest) | –0.03[b] (0.18) | 0.04[a] (0.19) | –0.11[a] (0.23) | 0.29*[a] (1.21) | – | | | |
| 6. Offset mean (rest) | –0.21[b] (0.49) | 0.08[a] (0.21) | –0.17[a] (0.35) | 0.25[a] (0.69) | 0.68***[c] (>10,000) | – | | |
| 7. Exponent left STS (rest) | –0.16[b] (0.32) | 0.01[a] (0.18) | –0.07[a] (0.20) | 0.35*[a] (2.87) | 0.68***[c] (>10,000) | 0.45***[c] (>10,000) | – | |
| 8. Offset left STS (rest) | –0.38**[b] (6.28) | –0.10[a] (0.23) | 0.02[a] (0.18) | 0.18[a] (0.38) | 0.18*[c] (0.80) | 0.47***[c] (>10,000) | 0.66***[c] (>10,000) | – |
| 9. Beta power left STS (rest) | –0.12[b] (0.25) | 0.17[a] (0.33) | –0.25[a] (0.74) | 0.01[a] (0.18) | 0.18*[c] (0.80) | 0.21*[c] (1.40) | 0.51***[c] (>10,000) | 0.59***[c] (>10,000) |
| *EEG language task* | | | | | | | | |
| 5. Exponent mean (task) | –0.10[b] (0.22) | 0.06[a] (0.20) | –0.15[a] (0.29) | 0.21[a] (0.49) | – | | | |
| 6. Offset mean (task) | –0.26[b] (0.92) | 0.09[a] (0.22) | –0.20[a] (0.43) | 0.22[a] (0.53) | 0.72***[c] (>10,000) | – | | |
| 7. Exponent left STS (task) | –0.20[b] (0.44) | 0.01[a] (0.18) | –0.09[a] (0.22) | 0.22[a] (0.51) | 0.65***[c] (>10000) | 0.51***[c] (>10,000) | – | |
| 8. Offset left STS (task) | –0.37**[b] (5.05) | –0.08[a] (0.21) | 0.00[a] (0.18) | 0.15[a] (0.30) | 0.29**[c] (18.31) | 0.55***[c] (>10,000) | 0.77***[c] (>10,000) | – |
| 9. Beta power left STS (task) | –0.19[b] (0.42) | 0.16[a] (0.32) | –0.22[a] (0.55) | –0.03[a] (0.19) | 0.05[c] (0.13) | 0.13[c] (0.31) | 0.47***[c] (>10,000) | 0.60***[c] (>10,000) |

Note: $r$ – Pearson's correlation coefficient; $BF_{10}$ – Bayes factor indicating ratio of the likelihood of an alternative hypothesis (H1) to a null hypothesis (H0); mean – values averaged across all electrodes; left STS – values averaged across three electrodes corresponding to the left superior temporal sulcus (T7, TP7, TP9).

***p < 0.001 (uncorrected); **p < 0.01 (uncorrected); *p < 0.05 (uncorrected)

[a] $n = 47$; [b] $n = 50$; [c] $n = 119$

ratio, and Glu/GABA+ imbalance the effect of GMV was regressed, while for exponents and offsets the effect of age was regressed. For zero-order correlations between variables, see *Appendix 1—table 3*.

Glu negatively correlated with offset in the left STS both at rest ($r$=–0.38, p=0.007, $BF_{10}$=6.28; *Figure 3A*) and during the language task ($r$=–0.37, p=0.009, $BF_{10}$=5.05), while any other correlations between Glu and EEG biomarkers were not significant and Bayesian statistics indicated in favor of null hypothesis or provided absence of evidence for either hypothesis. Furthermore, Glu/GABA+ imbalance positively correlated with exponent at rest both averaged across all electrodes ($r$=0.29, p=0.048, $BF_{10}$=1.21), as well as in the left STS electrodes ($r$=0.35, p=0.017, $BF_{10}$=2.87; *Figure 3B*) although Bayes factor provided absence of evidence for either alternative or null hypothesis. Conversely, GABA+ and Glu/GABA+ ratio were not significantly correlated with any of the EEG biomarkers and

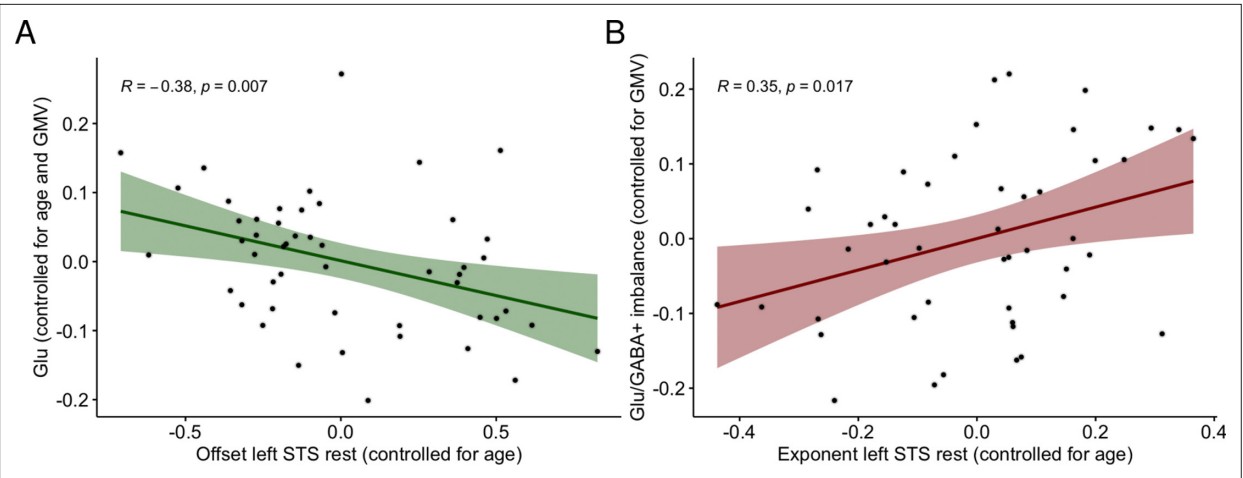

**Figure 3.** Relationships between electroencephalography (EEG) and magnetic resonance spectroscopy (MRS) excitatory and inhibitory (E/I) balance biomarkers. (**A**) Semi-partial correlation between offset at rest in the left superior temporal sulcus (STS) electrodes and Glu controlling for age and gray matter volume (GMV), n = 50 (DYS n = 21, CON n = 29). (**B**) Semi-partial correlation between exponent at rest in the left STS electrodes (controlled for age) and Glu/GABA+ imbalance (controlled for GMV), n = 47 (DYS n = 20, CON n = 27).

Bayesian statistics indicated in favor of null hypothesis or provided absence of evidence for either hypothesis.

## Testing the paths from neural noise to reading

The neural noise hypothesis of dyslexia predicts impact of the neural noise on reading through the impairment of (1) phonological awareness, (2) lexical access and generalization, and (3) multisensory integration (*Hancock et al., 2017*). Therefore, we analyzed correlations between these variables, reading skills and MRS and EEG biomarkers of E/I balance. For the composite score of phonological awareness, we averaged z-scores from phoneme deletion, phoneme and syllable spoonerisms tasks. For the composite score of lexical access and generalization we averaged z-scores from objects, colors, letters, and digits subtests from rapid automatized naming (RAN) task, while for the composite score of reading we averaged z-scores from words and pseudowords read per minute, and text reading time in reading comprehension task. The outcomes from the RAN and reading comprehension task have been transformed from raw time scores to items/time scores in order to provide the same direction of relationships for all z-scored measures, with greater values indicating better skills. For the multisensory integration score we used results from the redundant target effect task reported in our previous work (*Glica et al., 2024*), with greater values indicating a greater magnitude of multisensory integration.

Age positively correlated with multisensory integration (r=0.38, p<0.001, $BF_{10}$=87.98), composite scores of reading (r=0.22, p=0.014, $BF_{10}$=2.24), and phonological awareness (r=0.21, p=0.021, $BF_{10}$=1.59), while not with the composite score of RAN (r=0.13, p=0.151, $BF_{10}$=0.32). Hence, we regressed the effect of age from multisensory integration, reading, and phonological awareness scores and performed semi-partial correlations (*Table 3*, for zero-order correlations, see *Appendix 1—table 4*).

Phonological awareness positively correlated with offset in the left STS at rest (r=0.18, p=0.049, $BF_{10}$=0.77) and with beta power in the left STS both at rest (r=0.23, p=0.011, $BF_{10}$=2.73; *Figure 4A*) and during the language task (r=0.23, p=0.011, $BF_{10}$=2.84; *Figure 4B*), although Bayes factor provided absence of evidence for either alternative or null hypothesis. Furthermore, multisensory integration positively correlated with GABA+ concentration (r=0.31, p=0.034, $BF_{10}$=1.62) and negatively with Glu/GABA+ ratio (r=–0.32, p=0.029, $BF_{10}$=1.84), although Bayes factor provided absence of evidence for either alternative or null hypothesis. Any other correlations between reading skills and E/I balance biomarkers were not significant and Bayesian statistics indicated in favor of null hypothesis or provided absence of evidence for either hypothesis.

Given that beta power correlated with phonological awareness, and considering the prediction that neural noise impedes reading by affecting phonological awareness — we examined this relationship

**Table 3.** Semi-partial correlations between reading, phonological awareness, rapid automatized naming (RAN), multisensory integration, and biomarkers of excitatory-inhibitory balance.

For reading, phonological awareness, and multisensory integration the effect of age was regressed, for Glu the effects of age and gray matter volume (GMV) were regressed, for *GABA+, Glu/GABA+ ratio, and Glu/GABA+ imbalance the* effect of GMV was regressed, while for exponents and offsets the effect of age was regressed.

| Variable | 1. $r$ ($BF_{10}$) | 2. | 3. | 4. |
|---|---|---|---|---|
| *EEG resting state* | | | | |
| 1. Reading | – | | | |
| 2. Phonological awareness | 0.60***[c] (>10,000) | – | | |
| 3. RAN | 0.71***[c] (>10,000) | 0.48***[c] (>10,000) | – | |
| 4. Multisensory integration | 0.16[d] (0.41) | 0.25*[d] (2.09) | 0.02[d] (0.14) | – |
| 5. Glu | –0.03[b] (0.18) | –0.12[b] (0.25) | –0.05[b] (0.19) | 0.03[b] (0.18) |
| 6. GABA+ | –0.18[a] (0.36) | –0.06[a] (0.20) | –0.23[a] (0.56) | 0.31*[a] (1.62) |
| 7. Glu/GABA+ ratio | 0.07[a] (0.20) | 0.04[a] (0.19) | 0.09[a] (0.22) | –0.32*[a] (1.84) |
| 8. Glu/GABA+ imbalance | –0.21[a] (0.47) | –0.08[a] (0.21) | –0.20[a] (0.44) | 0.17[a] (0.33) |
| 9. Exponent mean (rest) | –0.08[c] (0.17) | 0.10[c] (0.20) | –0.06[c] (0.14) | 0.02[d] (0.14) |
| 10. Offset mean (rest) | 0.06[c] (0.14) | 0.14[c] (0.35) | 0.03[c] (0.12) | 0.16[d] (0.38) |
| 11. Exponent left STS (rest) | –0.08[c] (0.16) | 0.06[c] (0.14) | –0.04[c] (0.12) | –0.04[d] (0.14) |
| 12. Offset left STS (rest) | 0.12[c] (0.25) | 0.18*[c] (0.77) | 0.08[c] (0.17) | 0.08[d] (0.18) |
| 13. Beta power left STS (rest) | 0.04[c] (0.13) | 0.23*[c] (2.73) | –0.04[c] (0.12) | 0.05[d] (0.15) |
| *EEG language task* | | | | |
| 9. Exponent mean (task) | –0.07[c] (0.16) | 0.13[c] (0.30) | –0.10[c] (0.21) | 0.01[d] (0.14) |
| 10. Offset mean (task) | 0.05[c] (0.13) | 0.14[c] (0.34) | 0.01[c] (0.12) | 0.18[d] (0.50) |
| 11. Exponent left STS (task) | –0.03[c] (0.12) | 0.09[c] (0.18) | –0.07[c] (0.15) | –0.04[d] (0.14) |
| 12. Offset left STS (task) | 0.13[c] (0.28) | 0.18[c] (0.71) | 0.07[c] (0.15) | 0.09[d] (0.19) |
| 13. Beta power left STS (task) | 0.07[c] (0.15) | 0.23*[c] (2.84) | 0.02[c] (0.12) | 0.15[d] (0.33) |

Note: $r$ – Pearson's correlation coefficient; $BF_{10}$ – Bayes factor indicating ratio of the likelihood of an alternative hypothesis (H1) to a null hypothesis (H0); mean – values averaged across all electrodes; left STS – values averaged across three electrodes corresponding to the left superior temporal sulcus (T7, TP7, TP9).

***$p < 0.001$ (uncorrected); **$p < 0.01$ (uncorrected); *$p < 0.05$ (uncorrected)

[a]$n = 47$ [b]$n = 50$; [c]$n = 119$; [d]$n = 87$

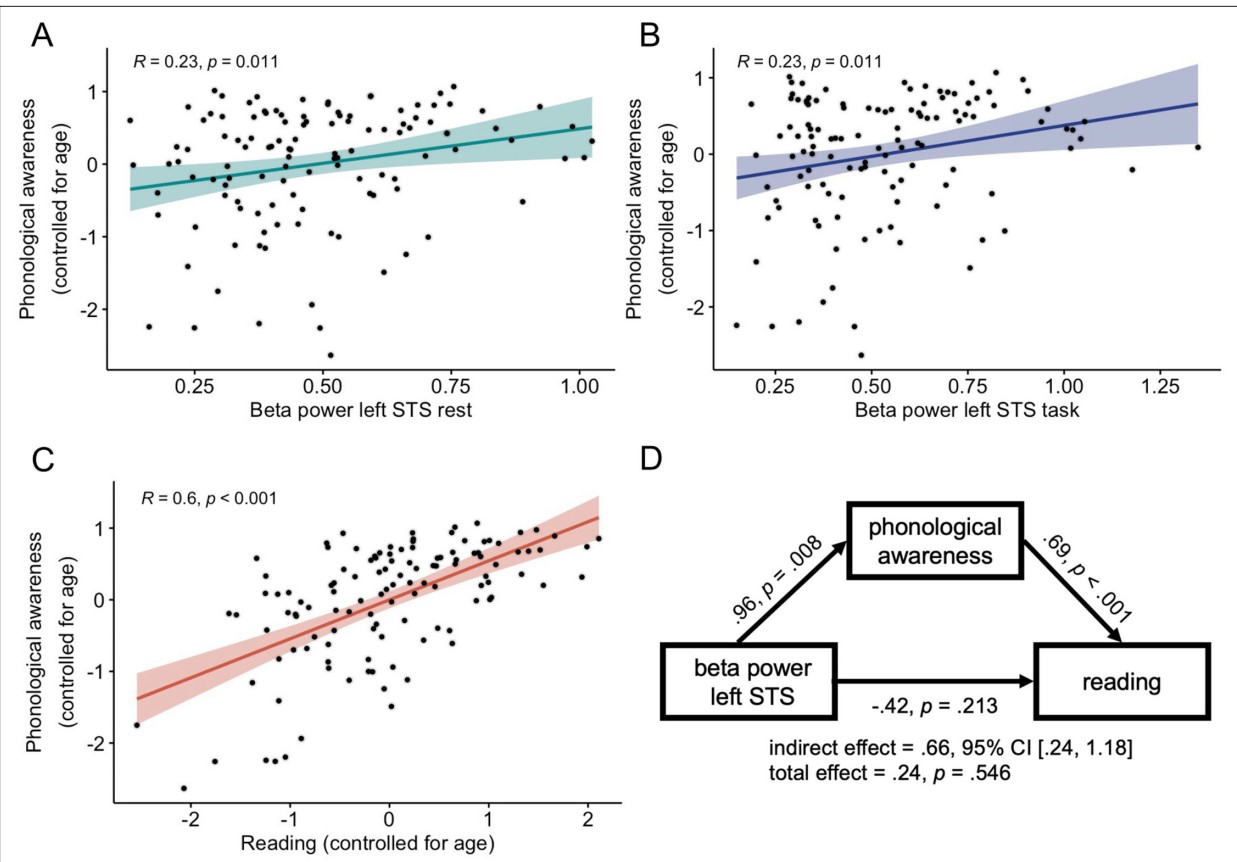

**Figure 4.** Associations between beta power, phonological awareness, and reading. $n$ = 119 (DYS $n$ = 59, CON $n$ = 60). (**A**) Semi-partial correlation between phonological awareness controlling for age and beta power (in the left superior temporal sulcus [STS] electrodes) at rest and (**B**) during the language task. (**C**) Partial correlation between phonological awareness and reading controlling for age. (**D**) Mediation analysis results. Unstandardized b regression coefficients are presented. Age was included in the analysis as a covariate. 95% CI – 95% confidence intervals; left STS – values averaged across three electrodes corresponding to the left superior temporal sulcus (T7, TP7, TP9).

through a mediation model. Since phonological awareness correlated with beta power in the left STS both at rest and during language task, the outcomes from these two conditions were averaged prior to the mediation analysis. Macro PROCESS v4.2 (**Hayes, 2017**) on IBM SPSS Statistics v29 with model 4 (simple mediation) with 5000 Bootstrap samples to assess the significance of indirect effect was employed. Since age correlated both with phonological awareness and reading, we also included age as a covariate.

The results indicated that both effects of beta power in the left STS ($b$=0.96, $t$(116) = 2.71, p=0.008, $BF_{incl}$ = 7.53) and age ($b$=0.06, $t$(116) = 2.55, p=0.012, $BF_{incl}$ = 5.98) on phonological awareness were significant. The effect of phonological awareness on reading was also significant ($b$=0.69, $t$(115) = 8.16, p<0.001, $BF_{incl}$>10,000), while the effects of beta power ($b$=–0.42, $t$(115) = –1.25, p=0.213, $BF_{incl}$ = 0.52) and age ($b$=0.03, $t$(115) = 1.18, p=0.241, $BF_{incl}$ = 0.49) on reading were not significant when controlling for phonological awareness. Finally, the indirect effect of beta power on reading through phonological awareness was significant ($b$=0.66, SE = 0.24, 95% CI = [0.24, 1.18]), while the total effect of beta power was not significant ($b$=0.24, $t$(116) = 0.61, p=0.546, $BF_{incl}$ = 0.41). The results from the mediation analysis are presented in **Figure 4D**.

Although similar mediation analysis could have been conducted for the Glu/GABA+ ratio, multi-sensory integration, and reading based on the correlations between these variables, we did not test this model due to the small sample size (47 participants), which resulted in insufficient statistical power.

## Discussion

The current study aimed to validate the neural noise hypothesis of dyslexia (**Hancock et al., 2017**) utilizing E/I balance biomarkers from EEG power spectra and ultra-high-field MRS. Contrary to its

predictions, we did not observe differences either in 1/$f$ slope, beta power, or Glu and GABA+ concentrations in participants with dyslexia. Relations between E/I balance biomarkers were limited to significant correlations between Glu and the offset when controlling for age, and between Glu/GABA+ imbalance and the exponent.

In terms of EEG biomarkers, our study found no evidence of group differences in the aperiodic components of the EEG signal. In most of the models, we did not find evidence for either including or excluding the effect of the group when Bayesian statistics were evaluated. The only exception was the regional analysis for the offset, where results indicated against including the group factor in the model. These findings diverge from previous research on an Italian cohort, which reported decreased exponent and offset in the dyslexic group at rest, specifically within the parieto-occipital region, but not the frontal region (*Turri et al., 2023*). Despite our study involving twice the number of participants and utilizing a longer acquisition time, we observed no group differences, even in the same cluster of electrodes (refer to Appendix 1). The participants in both studies were of similar ages. The only methodological difference – EEG acquisition with eyes open in our study versus both eyes-open and eyes-closed in the work by *Turri et al., 2023* – cannot fully account for the overall lack of group differences observed. The diverging study outcomes highlight the importance of considering potential inflation of effect sizes in studies with smaller samples.

Although a lower exponent of the EEG power spectrum has been associated with other neurodevelopmental disorders, such as ADHD (*Ostlund et al., 2021*) or ASD (but only in children with IQ below average) (*Manyukhina et al., 2022*), our study suggests that this is not the case for dyslexia. Considering the frequent comorbidity of dyslexia and ADHD (*Germanò et al., 2010*; *Langer et al., 2019*), increased neural noise could serve as a common underlying mechanism for both disorders. However, our specific exclusion of participants with a comorbid ADHD diagnosis indicates that the EEG spectral exponent cannot serve as a neurobiological marker for dyslexia in isolation. No information regarding such exclusion criteria was provided in the study by *Turri et al., 2023*; thus, potential comorbidity with ADHD may explain the positive findings related to dyslexia reported therein.

Regarding the aperiodic-adjusted oscillatory EEG activity, Bayesian statistics for beta power indicated in favor of excluding the group factor from the model. Non-significant group differences in beta power at rest have been previously reported in studies that did not account for aperiodic components (*Babiloni et al., 2012*; *Fraga González et al., 2018*; *Xue et al., 2020*). This again contrasts with the study by *Turri et al., 2023*, which observed lower aperiodic-adjusted beta power (at 15–25 Hz) in the dyslexic group. Concerning beta power during task, our results also contrast with previous studies which showed either reduced (*Spironelli et al., 2008*) or increased (*Rippon and Brunswick, 2000*) beta activity in participants with dyslexia. Nevertheless, since both of these studies employed phonological tasks involved children's samples and did not account for aperiodic activity, their relevance to our work is limited.

We could not perform analyses for the gamma oscillations since in the majority of participants the gamma peak was not detected above the aperiodic component. Due to the 1/$f$ properties of the EEG spectrum, both aperiodic and periodic components should be disentangled to analyze 'true' gamma oscillations; however, this approach is not typically recognized in electrophysiology research (*Hudson and Jones, 2022*). Indeed, previous studies that analyzed gamma activity in dyslexia (*Babiloni et al., 2012*; *Lasnick et al., 2023*; *Rufener and Zaehle, 2021*) did not separate the background aperiodic activity. For the same reason, we could not analyze results for the theta band, which often does not meet the criteria for an oscillatory component manifested as a peak in the power spectrum (*Klimesch, 1999*). Moreover, results from a study investigating developmental changes in both periodic and aperiodic components suggest that theta oscillations in older participants are mostly observed in frontal midline electrodes (*Cellier et al., 2021*), which were not analyzed in the current study.

Remarkably, in some models, results from Bayesian and frequentist statistics yielded divergent conclusions regarding the inclusion of non-significant effects. This was observed in more complex ANOVA models, whereas no such discrepancies appeared in $t$-tests or correlations. Given reports of high variability in Bayesian ANOVA estimates across repeated runs of the same analysis (*Pfister, 2021*), these results should be interpreted with caution. Therefore, following the recommendation to simplify complex models into Bayesian $t$-tests for more reliable estimates (*Pfister, 2021*), we conducted follow-up Bayesian $t$-tests in every case that favored the inclusion of non-significant interactions with the group factor. These analyses provided further evidence for the lack of differences

between the dyslexic and control groups. Another source of discrepancy between the two methods may stem from the inclusion of interactions between covariates and within-subject effects in frequentist ANOVA, which were not included in Bayesian ANOVA to adhere to the recommendation for simpler Bayesian models (*Pfister, 2021*).

In terms of neurometabolite concentrations derived from the MRS, we found no evidence for group differences in either Glu, GABA+, or Glu/GABA+ imbalance in the language-sensitive left STS. Conversely, the Bayes factor suggested against including the group factor in the model for the Glu/GABA+ ratio. While no previous study has localized the MRS voxel based on the individual activation levels, non-significant group differences in Glu and GABA concentrations within the temporo-parietal and visual cortices have been reported in both children and adults (*Kossowski et al., 2019*), as well as in the ACC in children (*Horowitz-Kraus et al., 2018*). Although our MRS sample size was half that of the EEG sample, previous research reporting group differences in Glu concentrations involved an even smaller dyslexic cohort (10 participants with dyslexia and 45 typical readers in *Pugh et al., 2014*). Consistent with earlier studies that identified group differences in Glu and GABA concentrations (*Del Tufo et al., 2018*; *Pugh et al., 2014*), we reported neurometabolite levels relative to tCr with GMV included as a covariate in the models, indicating that the absence of corresponding results cannot be ascribed to reference differences. Notably, our analysis of the fMRI localizer task revealed greater activation in the control group as compared to the dyslexic group within the left STS for words than control stimuli (see *Figure 2C* and *Appendix 1—table 5*, *Appendix 1—table 6*, and *Appendix 1—table 7*) in line with previous observations (*Blau et al., 2009*; *Dębska et al., 2021*; *Yan et al., 2021*).

We chose ultra-high-field MRS to improve data quality (*Özütemiz et al., 2023*) as the increased sensitivity and spectral resolution at 7T allows for better separation of overlapping metabolites compared to lower field strengths. Additionally, 7T provides a higher signal-to-noise ratio (SNR), improving the reliability of metabolite measurements and enabling the detection of small changes in Glu and GABA concentrations. Despite the theoretical advantages, several practical obstacles should be considered, such as susceptibility artifacts and inhomogeneities at higher field strengths that can impact data quality. Interestingly, actual methodological comparisons (*Pradhan et al., 2015*; *Terpstra et al., 2016*) show only a slight practical advantage of 7T single-voxel MRS when compared to optimized 3T acquisition. For example, fitting quality yielded reduced estimates of variance in concentration of Glu in 7T (CRLB) and slightly improved reproducibility levels for Glu and GABA (at both fields below 5%). Choosing the appropriate MRS sequence involves a trade-off between the accuracy of Glu and GABA measurements, as different sequences are recommended for each metabolite. J-edited MRS is recommended for measuring GABA, particularly with 3T scanners (*Mullins et al., 2014*). However, at 7T, more reliable results can be obtained using moderate echo-time, non-edited MRS (*Finkelman et al., 2022*). We have opted for a short-echo time sequence (28 ms), which is optimal for measuring Glu. However, this approach results in macromolecule contamination of the GABA signal (referred to as GABA+). Moreover, as *Finkelman et al., 2022*, demonstrated that TE of 80 ms was superior to a shorter echo time of 42 ms for resolving Glu and GABA at 7T, the data reported in the present study are likely influenced by the difficulty in resolving overlapping Glu and GABA peaks at 2.3 ppm, which could impact our measures of the Glu/GABA+ ratio and imbalance.

Irrespective of dyslexia status, we found negative correlations between age and exponent and offset, consistent with previous research (*Cellier et al., 2021*; *McSweeney et al., 2021*; *Schaworonkow and Voytek, 2021*; *Voytek et al., 2015*) and providing further evidence for maturational changes in the aperiodic components (indicative of increased E/I ratio). At the same time, in line with previous MRS works (*Kossowski et al., 2019*; *Marsman et al., 2013*), we observed a negative correlation between age and Glu concentrations. This suggests a contrasting pattern to EEG results, indicating a decrease in neuronal excitation with age. We also found a condition-dependent change in offset, with a lower offset observed at rest than during the language task. The offset value represents the uniform shift in power across frequencies (*Donoghue et al., 2020*), with a higher offset linked to increased neuronal spiking rates (*Manning et al., 2009*). Change in offset between conditions is consistent with observed increased alpha and beta power during the task, indicating elevated activity in both broadband (offset) and narrowband (alpha and beta oscillations) frequency ranges during the language task.

In regard to relationships between EEG and MRS E/I balance biomarkers, we observed a negative correlation between the offset in the left STS (both at rest and during the task) and Glu levels,

after controlling for age and GMV. This correlation was not observed in zero-order correlations (see *Appendix 1—table 3*). Contrary to our predictions, informed by previous studies linking the exponent to E/I ratio (*Colombo et al., 2019*; *Gao et al., 2017*; *Waschke et al., 2021*), we found the correlation with Glu levels to involve the offset rather than the exponent. This outcome was unexpected, as none of the referenced studies reported results for the offset. However, given the strong correlation between the exponent and offset observed in our study ($r=0.68$, $p<0.001$, $BF_{10}>10,000$ and $r=0.72$, $p<0.001$, $BF_{10}>10,000$ at rest and during the task respectively) it is conceivable that similar association might be identified for the offset if it were analyzed.

Nevertheless, previous studies examining relationships between EEG and MRS E/I balance biomarkers (*McKeon et al., 2024*; *van Bueren et al., 2023*) did not identify a similar negative association between Glu and the offset. Instead, one study noted a positive correlation between the Glu/GABA ratio and the exponent (*van Bueren et al., 2023*), which was significant in the intraparietal sulcus but not in the middle frontal gyrus. This finding presents counterintuitive evidence, suggesting that an increased E/I balance, as indicated by MRS, is associated with a higher aperiodic exponent, considered indicative of decreased E/I balance. In line with this pattern, another study discovered a positive relationship between the exponent and Glu levels in the dorsolateral prefrontal cortex (*McKeon et al., 2024*). Furthermore, they observed a positive correlation between the exponent and the Glu/GABA imbalance measure, calculated as the absolute residual value of a linear relationship between Glu and GABA (*McKeon et al., 2024*), a finding replicated in the current work. This implies that a higher spectral exponent might not be directly linked to MRS-derived Glu or GABA levels, but rather to a greater disproportion (in either direction) between these neurotransmitters. These findings, alongside the contrasting relationships between EEG and MRS biomarkers and age, suggest that these methods may reflect distinct biological mechanisms of E/I balance.

Evidence regarding associations between neurotransmitters levels and oscillatory activity also remains mixed. One study found a positive correlation between gamma peak frequency and GABA concentration in the visual cortex (*Muthukumaraswamy et al., 2009*), a finding later challenged by a study with a larger sample (*Cousijn et al., 2014*). Similarly, a different study noted a positive correlation between GABA in the left STS and gamma power (*Balz et al., 2016*), another study found non-significant relation between these measures (*Wyss et al., 2017*). Moreover, in a simultaneous EEG and MRS study, an event-related increase in Glu following visual stimulation was found to correlate with greater gamma power (*Lally et al., 2014*). We could not investigate such associations, as the algorithm failed to identify a gamma peak above the aperiodic component for the majority of participants. Also, contrary to previous findings showing associations between GABA in the motor and sensorimotor cortices and beta power (*Cheng et al., 2017*; *Gaetz et al., 2011*) or beta peak frequency (*Baumgarten et al., 2016*), we observed no correlation between Glu or GABA+ levels and beta power. However, these studies placed MRS voxels in motor regions which are typically linked to movement-related beta activity (*Baker et al., 1999*; *Rubino et al., 2006*; *Sanes and Donoghue, 1993*) and did not adjust beta power for aperiodic components, making direct comparisons with our findings limited.

Finally, we examined pathways posited by the neural noise hypothesis of dyslexia, through which increased neural noise may impact reading: phonological awareness, lexical access and generalization, and multisensory integration (*Hancock et al., 2017*). Phonological awareness was positively correlated with the offset in the left STS at rest, and with beta power in the left STS, both at rest and during the task. Additionally, multisensory integration showed correlations with GABA+ and the Glu/GABA+ ratio. Since the Bayes factor did not provide conclusive evidence supporting either the alternative or null hypothesis, these associations appear rather weak. Nonetheless, given the hypothesis's prediction of a causal link between these variables, we further examined a mediation model involving beta power, phonological awareness, and reading skills. The results suggested a positive indirect effect of beta power on reading via phonological awareness, whereas both the direct (controlling for phonological awareness and age) and total effects (without controlling for phonological awareness) were not significant. This finding is noteworthy, considering that participants with dyslexia exhibited reduced phonological awareness and reading skills, despite no observed differences in beta power. Given the cross-sectional nature of our study, further longitudinal research is necessary to confirm the causal relation among these variables. The effects of GABA+ and the Glu/GABA+ ratio on reading, mediated by multisensory integration, warrant further investigation. Additionally, considering our finding that only males with dyslexia showed deficits in multisensory integration (*Glica et al., 2024*),

sex should be considered as a potential moderating factor in future analyses. We did not test this model here due to the smaller sample size for GABA+ measurements.

Our findings suggest that the neural noise hypothesis, as proposed by *Hancock et al., 2017*, does not fully explain the reading difficulties observed in dyslexia. Despite the innovative use of both EEG and MRS biomarkers to assess E/I balance, neither method provided evidence supporting an E/I imbalance in dyslexic individuals. Importantly, our study focused on adolescents and young adults, and the EEG recordings were conducted during rest and a spoken language task. These factors may limit the generalizability of our results. Future research should include younger populations and incorporate a broader array of tasks, such as reading and phonological processing, to provide a more comprehensive evaluation of the E/I balance hypothesis. Moreover, since the MRS data was collected only from the left STS, it is plausible that other areas might be associated with differences in Glu or GABA concentrations in dyslexia. Nevertheless, the neural noise hypothesis predicted increased glutamatergic signaling in perisylvian regions, specifically in the left superior temporal cortex (*Hancock et al., 2017*). Furthermore, although our results do not support the idea of E/I balance alterations as a source of neural noise in dyslexia, they do not preclude other mechanisms leading to less synchronous neural firing posited by the hypothesis. In this context, there is evidence showing increased trial-to-trial inconsistency of neural responses in individuals with dyslexia (*Centanni et al., 2022*) or poor readers (*Hornickel and Kraus, 2013*) and its associations with specific dyslexia risk genes (*Centanni et al., 2018*; *Neef et al., 2017*). At the same time, the observed trial-to-trial inconsistency was either present only in a subset of participants (*Centanni et al., 2018*), limited to some experimental conditions (*Centanni et al., 2022*), or specific brain regions – e.g., brainstem in *Hornickel and Kraus, 2013*, left auditory cortex in *Centanni et al., 2018*, or left supramarginal gyrus in *Centanni et al., 2022*. Also, one study analyzing behavioral and fMRI response patterns did not find similar evidence for increased variability in dyslexia (*Tan et al., 2022*). Together, these results highlight the need to explore alternative neural mechanisms underlying dyslexia and suggest that cortical hyperexcitability may not be the primary cause of reading difficulties.

In conclusion, while our study challenges the neural noise hypothesis as a sole explanatory framework for dyslexia, it also underscores the complexity of the disorder and the necessity for multifaceted research approaches. By refining our understanding of the neural underpinnings of dyslexia, we can better inform future studies and develop more effective interventions for those affected by this condition.

## Materials and methods

### Participants

A total of 120 Polish participants aged between 15.09 and 24.95 years (*M*=19.47, *SD* = 3.06) took part in the study. This included 60 individuals with a clinical diagnosis of dyslexia performed by the psychological and pedagogical counseling centers (28 females and 32 males) and 60 control participants without a history of reading difficulties (28 females and 32 males). Since there are no standardized diagnostic norms for dyslexia in adults in Poland, individuals were assigned to the dyslexic group based on a past diagnosis of dyslexia. All participants were right-handed, born at term, without any reported neurological/psychiatric diagnosis and treatment (including ADHD), without hearing impairment, with normal or corrected-to-normal vision, and IQ higher than 80 as assessed by the Polish version of the Abbreviated Battery of the Stanford-Binet Intelligence Scale-Fifth Edition (SB5) (*Roid et al., 2017*).

The study was approved by the institutional review board at the University of Warsaw, Poland (reference number 2N/02/2021). All participants (or their parents in the case of underaged participants) provided written informed consent and received monetary remuneration for taking part in the study.

### Reading and reading-related tasks

Participants' reading skills were assessed by multiple paper-pencil tasks described in detail in our previous work (*Glica et al., 2024*). Briefly, we evaluated words and pseudowords read in 1 min (*Szczerbiński and Pelc Pękała, 2013*), RAN (*Fecenec et al., 2013*), and reading comprehension speed. We also assessed phonological awareness by a phoneme deletion task (*Szczerbiński and Pelc*

*Pękała, 2013*) and spoonerisms tasks (*Bogdanowicz et al., 2016*), as well as orthographic awareness (*Awramiuk and Krasowicz Kupis, 2014*). Furthermore, we evaluated non-verbal perception speed (*Ciechanowicz and Stańczak, 2006*) and short-term and working memory by forward and backward conditions from the Digit Span subtest from the WAIS-R (*Wechsler, 1981*). We also assessed participants' multisensory audiovisual integration by a redundant target effect task, which results have been reported in our previous work (*Glica et al., 2024*).

## EEG acquisition and procedure

EEG was recorded from 62 scalp and 2 ear electrodes using the Brain Products system (actiCHamp Plus, Brain Products GmbH, Gilching, Germany). Data were recorded in BrainVision Recorder Software (version 1.22.0002, Brain Products GmbH, Gilching, Germany) with a 500 Hz sampling rate. Electrodes were positioned in line with the extended 10–20 system. Electrode Cz served as an online reference, while the Fpz as a ground electrode. All electrodes' impedances were kept below 10 kΩ. Participants sat in a chair with their heads on a chin-rest in a dark, sound-attenuated, and electrically shielded room while the EEG was recorded during both a 5 min eyes-open resting state and the spoken language comprehension task. The paradigm was prepared in the Presentation software (version 20.1, Neurobehavioral Systems, Inc, Berkeley, CA, USA, https://www.neurobs.com/).

During rest, participants were instructed to relax and fixate their eyes on a white cross presented centrally on a black background. After 5 min, the spoken language comprehension task automatically started. The task consisted of 3–5 word-long sentences recorded in a speech synthesizer which were presented binaurally through sound-isolating earphones. After hearing a sentence, participants were asked to indicate whether the sentence was true or false by pressing a corresponding button. In total, there were 256 sentences – 128 true (e.g. 'Plants need water') and 128 false (e.g. 'Dogs can fly').

Sentences were presented in a random order in two blocks of 128 trials. At the beginning of each trial, a white fixation cross was presented centrally on a black background for 500 ms, then a blank screen appeared for either 500, 600, 700, or 800 ms (durations set randomly and equiprobably) followed by an auditory sentence presentation. The length of sentences ranged between 1.17 and 2.78 s and was balanced between true (*M*=1.82 s, *SD* = 0.29) and false sentences (*M*=1.82 s, *SD* = 0.32; $t(254)$ = –0.21, p=0.835; $BF_{10}$=0.14). After a sentence presentation, a blank screen was displayed for 1000 ms before starting the next trial. To reduce participants' fatigue, a 1 min break between two blocks of trials was introduced, and it took approximately 15 min to complete the task.

## fMRI acquisition and procedure

MRI data were acquired using Siemens 3T Trio system with a 32-channel head coil. Structural data were acquired using whole-brain 3D T1-weighted image (MP_RAGE, TI = 1100 ms, GRAPPA parallel imaging with acceleration factor PE = 2, voxel resolution = 1 mm$^3$, dimensions = 256×256×176). Functional data were acquired using whole-brain echo planar imaging sequence (TE = 30 ms, TR = 1410 ms, flip angle FA = 90°, FOV = 212 mm, matrix size = 92×92, 60 axial slices 2.3 mm thick, 2.3×2.3 mm$^2$ in-plane resolution, multiband acceleration factor = 3). Due to a technical issue, data from two participants were acquired with a 12-channel coil using whole-brain echo planar imaging sequence (TE = 28 ms, TR = 2500 ms, flip angle FA = 80°, FOV = 216 mm, matrix size = 72×72, 42 axial slices 3 mm thick, 3×3 mm$^2$ in-plane resolution).

The fMRI task served as a localizer for later MRS voxel placement in language-sensitive left STS. The task was prepared using Presentation software (version 20.1, Neurobehavioral Systems, Inc, Berkeley, CA, USA, https://www.neurobs.com/) and consisted of three runs, each lasting 5 min and 9 s. Two runs involved the presentation of visual stimuli, while the third run of auditory stimuli. In each run, stimuli were presented in 12 blocks, with 14 stimuli per block. In visual runs, there were four blocks from each category: (1) 3–4 letters-long words, (2) the same words presented as a false font string (BACS font) (*Vidal et al., 2017*), and (3) strings of 3–4-long consonants. Similarly, in the auditory run, there were four blocks from each category: (1) words recorded in a speech synthesizer, (2) the same words presented backward, and (3) consonant strings recorded in a speech synthesizer. Stimuli within each block were presented for 800 ms with a 400 ms break in between. The duration of each block was 16.8 s. Between blocks, a fixation cross was displayed for 8 s. Participants performed a 1-back task to maintain focus. The blocks were presented in a pseudorandom order and each block included 2–3 repeated stimuli.

## MRS acquisition and procedure

The GE 7T system with a 32-channel coil was utilized. Structural data were acquired using whole-brain 3D T1-weighted image (3D-SPGR BRAVO, TI = 450 ms, TE = 2.6 ms, TR = 6.6 ms, flip angle = 12°, bandwidth = ±32.5 kHz, ARC acceleration factor PE = 2, voxel resolution = 1 mm, dimensions = 256×256×180). MRS spectra with 320 averages were acquired from the left STS using single-voxel spectroscopy semiLaser sequence (*Deelchand et al., 2021*) (voxel size = 15×15×15 mm$^3$, TE = 28 ms, TR = 4000 ms, 4096 data points, water suppressed using VAPOR). Eight averages with unsuppressed water as a reference were collected.

To localize left STS, T1-weighted images from fMRI and MRS sessions were coregistered and fMRI peak coordinates were used as a center of voxel volume for MRS. Voxels were then adjusted to include only the brain tissue. During the acquisition, participants took part in a simple orthographic task.

## Statistical analyses
### EEG data

The continuous EEG signal was preprocessed in the EEGLAB (*Delorme and Makeig, 2004*). The data were filtered between 0.5 and 45 Hz (Butterworth filter, fourth order) and re-referenced to the average of both ear electrodes. The data recorded during the break between blocks, as well as bad channels, were manually rejected. The number of rejected channels ranged between 0 and 4 (*M*=0.19, *SD* = 0.63). Next, independent component analysis was applied. Components were automatically labeled by ICLabel (*Pion-Tonachini et al., 2019*), and those classified with 50–100% source probability as eye blinks, muscle activity, heart activity, channel noise, and line noise, or with 0–50% source probability as brain activity, were excluded. Components labeled as 'other' were visually inspected, and those identified as eye blinks and muscle activity were also rejected. The number of rejected components ranged between 11 and 46 (*M*=28.43, *SD* = 7.26). Previously rejected bad channels were interpolated using the nearest neighbor spline (*Perrin et al., 1987*; *Perrin et al., 1989*).

The preprocessed data were divided into a 5 min resting-state signal and a signal recorded during a spoken language comprehension task using MNE (*Gramfort, 2013*) and custom Python scripts. The signal from the task was cut up based on the event markers indicating the beginning and end of a sentence. Only trials with correct responses given between 0 and 1000 ms after the end of a sentence were included. The signals recorded during every trial were further multiplied by the Tukey window with $\alpha$=0.01 in order to normalize signal amplitudes at the beginning and end of every trial. This allowed a smooth concatenation of signals recorded during task trials, resulting in a continuous signal derived only when participants were listening to the sentences.

The continuous signal from the resting state and the language task was epoched into 2-s-long segments. An automatic rejection criterion of ±200 μV was applied to exclude epochs with excessive amplitudes. The number of epochs retained in the analysis ranged between 140 and 150 (*M*=149.66, *SD* = 1.20) in the resting-state condition and between 102 and 226 (*M*=178.24, *SD* = 28.94) in the spoken language comprehension task.

Power spectral density (PSD) for 0.5–45 Hz in 0.5 Hz increments was calculated for every artifact-free epoch using Welch's method for 2-s-long data segments windowed with a Hamming window with no overlap. The estimated PSDs were averaged for each participant and each channel separately for the resting-state condition and the language task. Aperiodic and periodic (oscillatory) components were parameterized using the FOOOF method (*Donoghue et al., 2020*). For each PSD, we extracted parameters for the 1–43 Hz frequency range using the following settings: *peak_width_limits* = [1, 12], *max_n_peaks* = infinite, *peak_threshold* = 2.0, *mean_peak_height* = 0.0, *aperiodic_mode* = 'fixed'.

Two broadband aperiodic parameters were extracted: the exponent, which quantifies the steepness of the EEG power spectrum, and the offset, which indicates signal's power across the entire frequency spectrum. We also extracted aperiodic-adjusted oscillatory power, bandwidth, and the center frequency parameters for the theta (4–7 Hz), alpha (7–14 Hz), beta (14–30 Hz), and gamma (30–43 Hz) bands. Since in the majority of participants, the algorithm did not find the peak above the aperiodic component in theta and gamma bands, we calculated the results only for the alpha and beta bands. The results for other periodic parameters than the beta power are reported in Appendix 1.

First, exponent and offset values were averaged across all electrodes and analyzed using a 2×2 repeated measures ANOVA with group (dyslexic, control) as a between-subjects factor and condition

(resting state, language task) as a within-subjects factor. Age was included in the analyses as a covariate due to correlation between variables.

Next, exponent and offset values were averaged across electrodes corresponding to the left (F7, FT7, FC5) and right inferior frontal gyrus (F8, FT8, FC6), and to the left (T7, TP7, TP9) and right superior temporal sulcus (T8, TP8, TP10). The electrodes were selected based on the analyses outlined by *Giacometti et al., 2014*, and *Scrivener and Reader, 2022*. For these analyses, a 2×2×2×2 repeated measures ANOVA with age as a covariate was conducted with group (dyslexic, control) as a between-subjects factor and condition (resting state, language task), hemisphere (left, right), and region (frontal, temporal) as within-subjects factors.

Results for the alpha and beta bands were calculated for the same clusters of frontal and temporal electrodes and analyzed with a similar 2×2×2×2 repeated measures ANOVA; however, for these analyses, age was not included as a covariate due to a lack of significant correlations.

Apart from the frequentist statistics, we also performed Bayesian statistics using JASP (*JASP Team, 2023*). For Bayesian repeated measures ANOVA, we reported the Bayes factor for the inclusion of a given effect ($BF_{incl}$) with the 'across matched model' option, as suggested by *Keysers et al., 2020*, calculated as a likelihood ratio of models with a presence of a specific factor to equivalent models differing only in the absence of the specific factor. For Bayesian *t*-tests and correlations, we reported the $BF_{10}$ value, indicating the ratio of the likelihood of an alternative hypothesis to a null hypothesis. We considered $BF_{incl/10} > 3$ and $BF_{incl/10} < 1/3$ as evidence for alternative and null hypotheses respectively, while $1/3 < BF_{incl/10} < 3$ as the absence of evidence (*Keysers et al., 2020*).

## fMRI data

### MRS voxel localization in the native space

The data were analyzed using Statistical Parametric Mapping (SPM12, Wellcome Trust Centre for Neuroimaging, London, UK) run on MATLAB R2020b (The MathWorks Inc, Natick, MA, USA). First, all functional images were realigned to the participant's mean. Then, T1-weighted images were coregistered to functional images for each subject. Finally, fMRI data were smoothed with a 6 mm isotropic Gaussian kernel.

In each subject, the left STS was localized in the native space as a cluster in the middle and posterior left superior temporal sulcus, exhibiting higher activation for visual words versus false font strings and auditory words versus backward words (logical AND conjunction) at p<0.01 uncorrected. For six participants (DYS *n*=2, CON *n*=4), the threshold was lowered to p<0.05 uncorrected, while for another six participants (DYS *n*=3, CON *n*=3) the contrast from the auditory run was changed to auditory words versus fixation cross due to a lack of activation for other contrasts.

### Group-level analysis

The group-level results from the fMRI task are reported in *Appendix 1—table 5*, *Appendix 1—table 6*, and *Appendix 1—table 7*. For this analysis, the following preprocessing steps were conducted: (1) realignment of all functional images to the participant's mean, (2) coregistration of T1-weighted images to functional images for each subject, (3) segmentation of coregistered anatomical images, (4) normalization of functional images, and (5) smoothing of functional data with a 6 mm isotropic Gaussian kernel. We also used ART (artifact detection tools with default options) to add movement regressors and reject volumes identified as motion outliers.

In the second-level analysis, we conducted one-sample *t*-tests to examine activation maps separately for visual (words>false fonts) and auditory runs (words>backward) within each group (dyslexic, control). We also employed paired *t*-tests to assess activations for both visual and auditory runs (logical AND conjunction), examining them separately for the groups. Finally, a flexible factorial model was utilized with the main factor of subject and an interaction factor between group and condition to investigate the effect of group in both visual and auditory runs. We reported the results at p<0.001 height threshold corrected for multiple comparisons using p<0.05 FWE cluster threshold. The anatomical regions were labeled based on the AAL3 atlas (*Rolls et al., 2020*). The results are reported for 50 participants – 21 in the dyslexic (12 females, 9 males) and 29 in the control group (13 females, 16 males) reflecting the sample size for Glu results from the MRS.

## MRS data

MRS data were analyzed using fsl-mrs version 2.0.7 (*Clarke et al., 2021*). Data stored in pfile format were converted into NIfTI-MRS using spec2nii tool. We then used the *fsl_mrs_preproc* function to automatically perform coil combination, frequency and phase alignment, bad average removal, combination of spectra, eddy current correction, shifting frequency to reference peak and phase correction.

To obtain information about the percentage of WM, GM, and CSF in the voxel, we used the *svs_segmentation* with results of *fsl_anat* as an input. Voxel segmentation was performed on structural images from a 3T scanner, coregistered to 7T structural images in SPM12, as the latter exhibited excessive artifacts and intensity bias in the temporal regions. Next, quantitative fitting was performed using *fsl_mrs* function. As a basis set, we utilized a collection of 27 metabolite spectra simulated using FID-A (*Simpson et al., 2017*) and a script tailored for our experiment. We supplemented this with synthetic macromolecule spectra provided by *fsl_mrs*. In contrast to symmetrical editing technique (*Edden et al., 2012*), we were unable to fully separate GABA peaks from the spectra of macromolecules. Therefore, we use 'GABA+' as an abbreviation to indicate the potential contributions of other compounds. Signals acquired with unsuppressed water served as water reference.

Spectra underwent quantitative assessment and visual inspection and those with linewidth higher than 20 Hz, %CRLB higher than 20%, and poor fit to the model were excluded from the analysis (see *Appendix 1—table 8* for a detailed checklist). Glu and GABA+ concentrations were expressed as a ratio to tCr (creatine+phosphocreatine) following previous MRS studies in dyslexia (*Del Tufo et al., 2018*; *Pugh et al., 2014*).

To analyze the metabolite results, separate univariate ANCOVAs were conducted for Glu, GABA+, Glu/GABA+ ratio, and Glu/GABA+ imbalance measures with group (control, dyslexic) as a between-subjects factor and voxel GMV as a covariate. Additionally, for the Glu analysis, age was included as a covariate due to a correlation between variables. Both frequentist and Bayesian statistics were calculated. Glu/GABA+ imbalance measure was calculated as the square root of the absolute residual value of a linear relationship between Glu and GABA+ (*McKeon et al., 2024*).

## Acknowledgements

This study was supported by the National Science Centre grant (2019/35/B/HS6/01763) awarded to Katarzyna Jednoróg. We gratefully acknowledge valuable discussions with Ralph Noeske from GE Healthcare for his support in setting up the protocol for an ultra-high-field MR spectroscopy and sharing the set-up for basis set simulation in FID-A.

## Additional information

### Funding

| Funder | Grant reference number | Author |
| --- | --- | --- |
| Narodowe Centrum Nauki | 2019/35/B/HS6/01763 | Katarzyna Jednoróg |

The funders had no role in study design, data collection and interpretation, or the decision to submit the work for publication.

### Author contributions

Agnieszka Glica, Katarzyna Wasilewska, Data curation, Formal analysis, Investigation, Visualization, Writing – original draft; Julia Jurkowska, Data curation, Investigation, Writing – review and editing; Jarosław Żygierewicz, Supervision, Methodology, Writing – review and editing; Bartosz Kossowski, Conceptualization, Supervision, Methodology, Writing – review and editing; Katarzyna Jednoróg, Conceptualization, Supervision, Funding acquisition, Visualization, Methodology, Writing – original draft, Project administration, Writing – review and editing

### Author ORCIDs

Agnieszka Glica ⬥ https://orcid.org/0000-0003-4102-8950
Katarzyna Jednoróg ⬥ https://orcid.org/0000-0003-3072-6956

## Ethics

The study was approved by the institutional review board at the University of Warsaw, Poland (reference number 2N/02/2021). All participants (or their parents in the case of underaged participants) provided written informed consent and received monetary remuneration for taking part in the study.

Reviewer #1 (Public review): https://doi.org/10.7554/eLife.99920.4.sa1
Reviewer #2 (Public review): https://doi.org/10.7554/eLife.99920.4.sa2
Reviewer #3 (Public review): https://doi.org/10.7554/eLife.99920.4.sa3
Author response https://doi.org/10.7554/eLife.99920.4.sa4

## Additional files

### Supplementary files
MDAR checklist

### Data availability

Behavioral data, raw and preprocessed EEG data, 2nd level fMRI data, preprocessed MRS data and Python script for the analysis of preprocessed EEG data can be found at OSF: https://osf.io/4e7ps/ (https://doi.org/10.17605/OSF.IO/4E7PS).

The following dataset was generated:

| Author(s) | Year | Dataset title | Dataset URL | Database and Identifier |
| --- | --- | --- | --- | --- |
| Glica A, Wasilewska K | 2024 | Neural Noise | https://doi.org/10.17605/OSF.IO/4E7PS | Open Science Framework, 10.17605/OSF.IO/4E7PS |

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

## Appendix 1

**Appendix 1—table 1.** Descriptive statistics for electroencephalography (EEG) and magnetic resonance spectroscopy (MRS) results separately for the groups.

| | DYS | | CON | |
|---|---|---|---|---|
| | *M* | *SD* | *M* | *SD* |
| EEG resting state[a] | | | | |
| Exponent mean (rest) | 1.54 | 0.14 | 1.54 | 0.18 |
| Exponent left IFG (rest) | 1.54 | 0.16 | 1.53 | 0.18 |
| Exponent left STS (rest) | 1.50 | 0.18 | 1.47 | 0.22 |
| Exponent right IFG (rest) | 1.54 | 0.15 | 1.54 | 0.18 |
| Exponent right STS (rest) | 1.48 | 0.18 | 1.45 | 0.22 |
| Offset mean (rest) | −10.80 | 0.19 | −10.80 | 0.24 |
| Offset left IFG (rest) | −10.72 | 0.34 | −10.74 | 0.33 |
| Offset left STS (rest) | −10.97 | 0.38 | −10.98 | 0.37 |
| Offset right IFG (rest) | −10.79 | 0.29 | −10.81 | 0.32 |
| Offset right STS (rest) | −10.99 | 0.31 | −11.04 | 0.36 |
| Beta power left IFG (rest) | 0.48 | 0.18 | 0.48 | 0.20 |
| Beta power left STS (rest) | 0.49 | 0.19 | 0.48 | 0.21 |
| Beta power right IFG (rest) | 0.49 | 0.18 | 0.50 | 0.19 |
| Beta power right STS (rest) | 0.51 | 0.20 | 0.50 | 0.21 |
| EEG language task[a] | | | | |
| Exponent mean (task) | 1.55 | 0.15 | 1.56 | 0.18 |
| Exponent left IFG (task) | 1.55 | 0.16 | 1.55 | 0.19 |
| Exponent left STS (task) | 1.50 | 0.20 | 1.47 | 0.21 |
| Exponent right IFG (task) | 1.54 | 0.17 | 1.55 | 0.19 |
| Exponent right STS (task) | 1.47 | 0.19 | 1.45 | 0.22 |
| Offset mean (task) | −10.67 | 0.25 | −10.67 | 0.28 |
| Offset left IFG (task) | −10.58 | 0.39 | −10.60 | 0.37 |
| Offset left STS (task) | −10.86 | 0.44 | −10.87 | 0.42 |
| Offset right IFG (task) | −10.65 | 0.36 | −10.68 | 0.37 |
| Offset right STS (task) | −10.88 | 0.36 | −10.94 | 0.41 |
| Beta power left IFG (task)[b] | 0.50 | 0.23 | 0.51 | 0.21 |
| Beta power left STS (task) | 0.54 | 0.24 | 0.53 | 0.23 |
| Beta power right IFG (task) | 0.51 | 0.23 | 0.52 | 0.21 |
| Beta power right STS (task) | 0.55 | 0.26 | 0.55 | 0.23 |
| *MRS* | | | | |
| Glu[c] | 1.11 | 0.12 | 1.07 | 0.10 |
| GABA+[d] | 0.46 | 0.14 | 0.44 | 0.12 |
| Glu/GABA +ratio[d] | 2.67 | 0.87 | 2.68 | 0.75 |
| Glu/GABA +imbalance[d] | 0.27 | 0.11 | 0.24 | 0.12 |

*Appendix 1—table 1 Continued on next page*

*Appendix 1—table 1 Continued*

|  | DYS | CON |
| --- | --- | --- |

Note: DYS – dyslexic group; CON – control group; mean – values averaged across all electrodes; left IFG – values averaged across three electrodes corresponding to the left inferior frontal gyrus (F7, FT7, FC5); left STS – values averaged across three electrodes corresponding to the left superior temporal sulcus (T7, TP7, TP9); right IFG – values averaged across three electrodes corresponding to the right inferior frontal gyrus (F8, FT8, FC6); right STS – values averaged across three electrodes corresponding to the right superior temporal sulcus (T8, TP8, TP10).

[a] n = 119 (DYS n = 59, CON n = 60); [b] n = 117 (DYS n = 57, CON n = 60); [c] n = 50 (DYS n = 21, CON n = 29); [d] n = 47 (DYS n = 20, CON n = 27).

## EEG results – frontal and temporal electrodes

### Beta (14–30 Hz) aperiodic-adjusted

### Beta center frequency

There was a significant effect of condition ($F(1,115) = 6.12$, p=0.015, $\eta^2_p = 0.051$, $BF_{incl} = 2.94$) with post hoc comparison indicating that the beta peak was at higher frequencies at rest ($M$=19.86, $SD = 2.64$) than during the language task ($M$=19.44, $SD = 2.48$, $p_{corr} = 0.015$). There was also a significant interaction between condition and region, although the Bayes factor did not provide evidence for either inclusion or exclusion ($F(1,115) = 5.96$, p=0.016, $\eta^2_p = 0.049$, $BF_{incl} = 1.52$). Post hoc comparisons indicated that the beta peak was at higher frequencies at rest than during the language task in the frontal region ($M_{rest} = 20.03$, $SD_{rest} = 2.79$, $M_{task} = 19.43$, $SD_{task} = 2.51$, $p_{corr} = 0.002$), while this difference was not significant in the temporal region ($M_{rest} = 19.68$, $SD_{rest} = 2.76$, $M_{task} = 19.45$, $SD_{task} = 2.70$, $p_{corr} = 0.207$). Furthermore, at rest, the beta peak was at higher frequencies in the frontal as compared to the temporal region ($p_{corr} = 0.028$), while this difference was not significant during the language task ($p_{corr} = 0.878$). The effect of group ($F(1,115) = 0.02$, p=0.896, $\eta^2_p = 0.000$, $BF_{incl} = 0.001$) was not significant and Bayes factor indicated against including it in the model. Any other effects of interactions were not significant and Bayesian statistics indicated against including these factors in the model or did not provide evidence for either inclusion or exclusion.

### Beta bandwidth

The interaction between region, hemisphere, and group was not significant, although Bayesian statistics indicated in favor of including it in the model ($F(1,115) = 1.92$, p=0.169, $\eta^2_p = 0.016$, $BF_{incl} = 389.67$). The effect of group ($F(1,115) = 0.39$, p=0.532, $\eta^2_p = 0.003$, $BF_{incl} = 0.60$) was not significant while Bayes factor did not provide evidence for either inclusion or exclusion. Any other effects of interactions were not significant and Bayesian statistics indicated against including them in the model or did not provide evidence for either inclusion or exclusion. Since Bayes factor suggested the inclusion of the region*hemisphere*group interaction in the model, we further conducted Bayesian *t*-tests to determine whether this was driven by differences between control and dyslexic groups. The results, however, supported the null hypothesis in both the left ($M_{DYS} = 7.19$, $SD_{DYS} = 2.64$, $M_{CON} = 6.96$, $SD_{CON} = 2.84$, $BF_{10}$=0.22) and right hemisphere in the frontal region ($M_{DYS} = 6.93$, $SD_{DYS} = 2.86$, $M_{CON} = 7.07$, $SD_{CON} = 2.80$, $BF_{10}$=0.20) as well as in the left hemisphere in the temporal region ($M_{DYS} = 7.32$, $SD_{DYS} = 2.57$, $M_{CON} = 6.86$, $SD_{CON} = 2.72$, $BF_{10}$=0.29). The results in the right hemisphere in the temporal region were inconclusive ($M_{DYS} = 7.09$, $SD_{DYS} = 2.25$, $M_{CON} = 6.36$, $SD_{CON} = 2.61$, $BF_{10}$=0.66).

### Alpha (7–14 Hz) aperiodic-adjusted

For these analyses, the sample size was 112 (DYS *n*=56, CON *n*=56), since in 7 participants the algorithm did not find the alpha peak above the aperiodic component in selected electrodes.

### Alpha power

There was a significant effect of condition ($F(1,110) = 63.47$, p<0.001, $\eta^2_p = 0.366$, $BF_{incl}$>10,000) with post hoc comparison indicating that the alpha power was greater during the language task ($M$=1.21, $SD = 0.47$) than at rest ($M$=0.99, $SD = 0.39$, $p_{corr}$<0.001). There were also significant effects of hemisphere ($F(1,110) = 13.84$, p<0.001, $\eta^2_p = 0.112$, $BF_{incl} = 76.81$) and region ($F(1,110) = 6.34$, p=0.013, $\eta^2_p = 0.054$, $BF_{incl} = 2.98$). For the main effect of hemisphere, post hoc comparison indicated that alpha power was greater in the right ($M$=1.11, $SD = 0.41$) as compared to the left hemisphere ($M$=1.09, $SD = 0.42$, $p_{corr}$<0.001), while for the main effect of region, post hoc comparison indicated that the alpha power was greater in the temporal ($M$=1.11, $SD = 0.41$) as compared to the frontal region ($M$=1.09, $SD = 0.42$, $p_{corr} = 0.013$). Furthermore, there were significant interactions between

condition, region, and group ($F(1,110)$ = 4.78, p=0.031, $\eta^2_p$ = 0.042, $BF_{incl}$ = 64.84) as well as between hemisphere and region ($F(1,110)$ = 4.35, p=0.039, $\eta^2_p$ = 0.038, $BF_{incl}$ = 0.92) although Bayes factor did not provide evidence for either inclusion or exclusion the hemisphere*region interaction. For the condition*region*group interaction, post hoc comparisons indicated that in both groups and in both regions, alpha power was greater in the language task than at rest (for all comparisons $p_{corr}$<0.001). Furthermore, in the control group during resting-state condition, alpha power was greater in the temporal ($M$=0.99, $SD$ = 0.36) as compared to the frontal region ($M$=0.95, $SD$ = 0.38, $p_{corr}$=0.003), while any other comparisons were not significant. For the hemisphere*region interaction, post hoc comparisons indicated that greater alpha power in the temporal as compared to the frontal region was significant in the right ($M_{frontal}$ = 1.10, $SD_{frontal}$ = 0.42, $M_{temporal}$ = 1.13, $SD_{temporal}$ = 0.40, $p_{corr}$ = 0.001), while not in the left hemisphere ($M_{frontal}$ = 1.08, $SD_{frontal}$ = 0.42, $M_{temporal}$ = 1.09, $SD_{temporal}$ = 0.42, $p_{corr}$ = 0.386). Also, in the temporal region, greater alpha power was found in the right than in the left hemisphere ($p_{corr}$<0.001), while the difference between hemispheres was not significant in the frontal region ($p_{corr}$ = 0.110). The effect of group ($F(1,110)$ = 0.27, p=0.607, $\eta^2_p$ = 0.002, $BF_{incl}$ = 0.02) was not significant and Bayes factor indicated against including it in the model. Any other interactions were not significant and Bayesian statistics indicated against including them in the model or did not provide evidence for either inclusion or exclusion.

## Alpha center frequency

There was a significant effect of condition ($F(1,110)$ = 15.24, p<0.001, $\eta^2_p$ = 0.122, $BF_{incl}$ = 144.27) with post hoc comparison indicating that alpha peak was at lower frequencies at rest ($M$=10.51, $SD$ = 0.98) than during the language task ($M$=10.73, $SD$ = 0.94, $p_{corr}$<0.001). There were also significant interactions between condition and hemisphere ($F(1,110)$ = 9.99, p=0.002, $\eta^2_p$ = 0.083, $BF_{incl}$ = 14.42), as well as between condition and region ($F(1,110)$ = 4.28, p=0.041, $\eta^2_p$ = 0.037, $BF_{incl}$ = 0.82), although Bayes factor did not provide evidence for either including or excluding condition*region interaction. For the condition*hemisphere interaction, post hoc comparisons indicated that the alpha peak was at lower frequencies at rest than during the language task both in the left ($M_{rest}$ = 10.59, $SD_{rest}$ = 1.04, $M_{task}$ = 10.72, $SD_{task}$ = 0.95, $p_{corr}$ = 0.048) and in the right hemisphere ($M_{rest}$ = 10.42, $SD_{rest}$ = 1.03, $M_{task}$ = 10.75, $SD_{task}$ = 0.95, $p_{corr}$<0.001). Furthermore, in the resting-state condition, the alpha peak was at lower frequencies in the right as compared to the left hemisphere ($p_{corr}$ = 0.008), while the difference between hemispheres was not significant during the language task ($p_{corr}$ = 0.334). For the condition*region interaction, post hoc comparisons indicated that alpha peak was at lower frequencies at rest than during the language task both in the temporal ($M_{rest}$ = 10.48, $SD_{rest}$ = 0.98, $M_{task}$ = 10.76, $SD_{task}$ = 0.96, $p_{corr}$ <0.001) and in the frontal region ($M_{rest}$ = 10.53, $SD_{rest}$ = 1.02, $M_{task}$ = 10.71, $SD_{task}$ = 0.97, $p_{corr}$ = 0.008), while the difference between frontal and temporal regions was not significant either at rest ($p_{corr}$ = 0.128) or during the language task ($p_{corr}$ = 0.288). The effect of group ($F(1,110)$ = 1.55, p=0.216, $\eta^2_p$ = 0.014, $BF_{incl}$ = 0.70) was not significant while Bayes factor did not provide evidence for either inclusion or exclusion. Any other interactions were not significant and Bayesian statistics indicated against including them in the model or did not provide evidence for either inclusion or exclusion.

## Alpha bandwidth

The analysis revealed a significant effect of condition ($F(1,110)$ = 6.21, p=0.014, $\eta^2_p$ = 0.053, $BF_{incl}$ = 3.06) with post hoc comparison indicating that the alpha peak was wider at rest ($M$=3.18, $SD$ = 1.25) than during the language task ($M$=2.91, $SD$ = 0.94, $p_{corr}$ = 0.014). There was also a significant effect of region, although Bayesian statistics did not provide evidence for either inclusion or exclusion ($F(1,110)$ = 5.42, p=0.022, $\eta^2_p$ = 0.047, $BF_{incl}$ = 1.64). Post hoc comparison indicated that the alpha peak was wider in the temporal ($M$=3.12, $SD$ = 0.94) as compared to the frontal region ($M$=2.97, $SD$ = 1.04, $p_{corr}$ = 0.022). There were also significant interactions between region and condition ($F(1,110)$ = 7.33, p=0.008, $\eta^2_p$ = 0.062, $BF_{incl}$ = 4.15) as well as between region and group ($F(1,110)$ = 5.59, p=0.020, $\eta^2_p$ = 0.048, $BF_{incl}$ = 0.38), although Bayes factor did not provide evidence for either including or excluding region*group interaction. For the region*condition interaction, post hoc comparisons indicated that the alpha peak was wider in the resting-state condition as compared to the language task in the frontal region ($M_{rest}$ = 3.18, $SD_{rest}$ = 1.45, $M_{task}$ = 2.77, $SD_{task}$ = 0.99, $p_{corr}$ = 0.001), while this difference was not significant in the temporal region ($M_{rest}$ = 3.19, $SD_{rest}$ = 1.22, $M_{task}$ = 3.04, $SD_{task}$ = 1.03, $p_{corr}$ = 0.225). Furthermore, during the language task, the alpha peak was wider in the temporal than in the frontal region ($p_{corr}$<0.001), while the difference between regions was not significant during the resting-state condition ($p_{corr}$ = 0.932). For the region*group interaction, post hoc comparisons indicated that in the dyslexic group, the alpha peak was wider in the temporal as compared to the frontal region ($M_{frontal}$ = 2.85, $SD_{frontal}$ = 1.09, $M_{temporal}$ = 3.14, $SD_{temporal}$ = 1.01,

$p_{corr}$ = 0.001), while this difference was not significant in the control group ($M_{frontal}$ = 3.10, $SD_{frontal}$ = 0.98, $M_{temporal}$ = 3.09, $SD_{temporal}$ = 0.87, $p_{corr}$ = 0.980). The difference between dyslexic and control groups was not significant either in the frontal ($p_{corr}$ = 0.214) or in the temporal region ($p_{corr}$ = 0.810). The interaction between region, hemisphere, and condition was not significant, although Bayesian statistics indicated in favor of including it in the model ($F(1,110)$ = 1.54, p=0.217, $\eta^2_p$ = 0.014, $BF_{incl}$ = 5.96). The effect of group ($F(1,110)$ = 0.33, p=0.569, $\eta^2_p$ = 0.003, $BF_{incl}$ = 0.05) was not significant and Bayes factor indicated against including it in the model. Any other interactions were not significant and Bayesian statistics indicated against including them in the model or did not provide evidence for either inclusion or exclusion.

## EEG results – parieto-occipital electrodes

Following the previous study, which revealed differences in aperiodic and periodic components between dyslexic and control groups in the parieto-occipital region (*Turri et al., 2023*), we conducted additional analyses using the same cluster of electrodes from the left (PO7, PO3, O1) and the right hemisphere (PO8, PO4, O2). For the exponent and offset, we employed a 2×2×2 (group, condition, hemisphere) repeated measures ANOVA with age included as a covariate. For the beta and alpha bands results, we used a similar model but without the effect of age included as a covariate.

### Exponent

The analysis revealed significant effects of age ($F(1,116)$ = 5.22, p=0.024, $\eta^2_p$ = 0.043, $BF_{incl}$ = 2.07) and hemisphere ($F(1,116)$ = 6.37, p=0.013, $\eta^2_p$ = 0.052, $BF_{incl}$>10,000) with post hoc comparison indicating that the exponent was lower in the left (M=1.46, SD = 0.21) as compared to the right hemisphere (M=1.53, SD = 0.19, $p_{corr}$<0.001). The effect of group was not significant ($F(1,116)$ = 0.07, p=0.786, $\eta^2_p$ = 0.001, $BF_{incl}$ = 0.65) although Bayes factor did not provide evidence for either inclusion or exclusion. Any other effects or interactions were not significant and Bayesian statistics indicated against including them in the model or did not provide evidence for either inclusion or exclusion.

### Offset

There were significant effects of hemisphere ($F(1,116)$ = 15.20, p<0.001, $\eta^2_p$ = 0.116, $BF_{incl}$>10,000) and condition ($F(1,116)$ = 8.70, p=0.004, $\eta^2_p$ = 0.070, $BF_{incl}$>10,000) with post hoc comparisons indicating that the offset was lower in the left (M=−11.19, SD = 0.52) as compared to the right hemisphere (M=−10.73, SD = 0.27, $p_{corr}$<0.001), and at rest (M=−11.03, SD = 0.35) than during the language task (M=−10.90, SD = 0.39, $p_{corr}$<0.001). The interaction between condition and hemisphere ($F(1,116)$ = 0.13, p=0.725, $\eta^2_p$ = 0.001, $BF_{incl}$ = 31.62) was not significant although Bayes factor indicated in favor of including it in the model. The effect of group ($F(1,116)$ = 0.08, p=0.781, $\eta^2_p$ = 0.001, $BF_{incl}$ = 0.04) was not significant and Bayes factor indicated against including it in the model. Any other effects or interactions were not significant and Bayesian statistics indicated against including them in the model or did not provide evidence for either inclusion or exclusion.

### Beta (14–30 Hz) aperiodic-adjusted

#### Beta power

The analysis revealed significant effects of hemisphere ($F(1,117)$ = 18.74, p<0.001, $\eta^2_p$ = 0.138, $BF_{incl}$ = 612.30) and condition ($F(1,117)$ = 24.05, p<0.001, $\eta^2_p$ = 0.170, $BF_{incl}$ = 4545.40). For the main effect of hemisphere, post hoc comparison indicated that the beta power was greater in the right (M=0.56, SD = 0.19) as compared to the left hemisphere (M=0.53, SD = 0.18, $p_{corr}$<0.001), while for the main effect of condition, post hoc comparison indicated that the beta power was greater during the language task (M=0.57, SD = 0.21) than at rest (M=0.51, SD = 0.18, $p_{corr}$<0.001). The effect of group was not significant ($F(1,117)$ = 0.06, p=0.841, $\eta^2_p$ = 0.000, $BF_{incl}$ = 0.55), although Bayes factor did not provide evidence for either inclusion or exclusion. Any other interactions were not significant and Bayesian statistics indicated against including them in the model or did not provide evidence for either inclusion or exclusion.

#### Beta center frequency

The analysis revealed significant interactions between group and hemisphere ($F(1,117)$ = 5.10, p=0.026, $\eta^2_p$ = 0.042, $BF_{incl}$ = 1.74), as well as between group, hemisphere, and condition ($F(1,117)$ = 4.15, p=0.044, $\eta^2_p$ = 0.034, $BF_{incl}$ = 1.89), although Bayes factor did not provide evidence for either inclusion or exclusion. For the group*hemisphere interaction, post hoc comparisons did not reveal

any significant differences. For the group*hemisphere*condition interaction, post hoc comparisons indicated that in the dyslexic group during the resting-state condition, beta peak was at lower frequencies in the right (*M*=18.51, *SD* = 1.95) as compared to the left hemisphere (*M*=19.07, *SD* = 2.24, $p_{corr}$ = 0.026), while any other comparisons were not significant. The effect of group was not significant (*F*(1,117) = 0.20, p = 0.659, $\eta^2_p$ = 0.002, $BF_{incl}$ = 0.37), although Bayes factor did not provide evidence for either inclusion or exclusion. Any other effects or interactions were not significant and Bayesian statistics indicated against including them in the model.

### Beta bandwidth

The effect of group was not significant (*F*(1,117) = 0.02, p=0.890, $\eta^2_p$ = 0.000, $BF_{incl}$ = 0.19) and Bayes factor indicated against including it in the model. Any other effects or interactions were not significant and Bayesian statistics indicated against including them in the model or did not provide evidence for either inclusion or exclusion.

## Alpha (7–14 Hz) aperiodic-adjusted

For these analyses, the sample size was 117 (DYS *n*=59, CON *n*=58), since in two participants the algorithm did not find the alpha peak above the aperiodic component in selected electrodes.

### Alpha power

There were significant effects of hemisphere (*F*(1,115) = 63.01, p<0.001, $\eta^2_p$ = 0.354, $BF_{incl}$>10,000) and condition (*F*(1,115) = 93.58, p<0.001, $\eta^2_p$ = 0.449, $BF_{incl}$>10,000). For the main effect of hemisphere, post hoc comparison indicated that the alpha power was greater in the right (*M*=1.30, *SD* = 0.36) as compared to the left hemisphere (*M*=1.22, *SD* = 0.34, $p_{corr}$<0.001), while for the main effect of condition, post hoc comparison indicated that the alpha power was greater during the language task (*M*=1.38, *SD* = 0.36) than at rest (*M*=1.15, *SD* = 0.38, $p_{corr}$<0.001). There were also significant interactions between group and hemisphere (*F*(1,115) = 5.25, p=0.024, $\eta^2_p$ = 0.044, $BF_{incl}$ = 2.26), as well as between hemisphere and condition (*F*(1,115) = 4.01, p=0.048, $\eta^2_p$ = 0.034, $BF_{incl}$ = 1.36), however Bayes factor did not provide the evidence for either inclusion or exclusion. For the group*hemisphere interactions, post hoc comparisons indicated that greater alpha power was in the right as compared to the left hemisphere both in the dyslexic ($M_{left}$ = 1.20, $SD_{left}$ = 0.35, $M_{right}$ = 1.31, $SD_{right}$ = 0.36, $p_{corr}$<0.001) and in the control group ($M_{left}$ = 1.24, $SD_{left}$ = 0.33, $M_{right}$ = 1.30, $SD_{right}$ = 0.36, $p_{corr}$<0.001), while the difference between the dyslexic and control group was not significant either in the left ($p_{corr}$ = 0.497) or in the right hemisphere ($p_{corr}$ = 0.926). For the hemisphere*condition interaction, post hoc comparisons indicated that the alpha power was greater in the right as compared to the left hemisphere both at rest and during the language task (all comparisons $p_{corr}$<0.001), and that the alpha power was greater during the task than at rest both in the left and in the right hemisphere (all comparisons $p_{corr}$<0.001). The effect of group was not significant (*F*(1,115) = 0.08, p=0.776, $\eta^2_p$ = 0.001, $BF_{incl}$ = 0.56), although Bayes factor did not provide evidence for either inclusion or exclusion. Any other interactions were not significant and Bayesian statistics indicated against including them in the model or did not provide evidence for either inclusion or exclusion.

### Alpha center frequency

There was a significant effect of condition (*F*(1,115) = 92.36, p<0.001, $\eta^2_p$ = 0.445, $BF_{incl}$>10,000) with post hoc comparison indicating that the alpha peak was at lower frequencies at rest (*M*=10.44, *SD* = 0.95) than during the language task (*M*=10.87, *SD* = 0.93, $p_{corr}$<0.001). The effect of group was not significant (*F*(1,115) = 2.94, p=0.089, $\eta^2_p$ = 0.025, $BF_{incl}$ = 0.60), although Bayes factor did not provide evidence for either inclusion or exclusion. Any other effects or interactions were not significant and Bayesian statistics indicated against including them in the model or did not provide evidence for either inclusion or exclusion.

### Alpha bandwidth

The effect of group was not significant (*F*(1,115) = 0.01, p=0.923, $\eta^2_p$ = 0.000, $BF_{incl}$ = 0.36), although Bayes factor did not provide evidence for either inclusion or exclusion. Any other effects or interactions were not significant and Bayesian statistics indicated against including them in the model or did not provide evidence for either inclusion or exclusion.

**Appendix 1—table 2.** Demographic and behavioral characteristics of the subsample of 47 participants.

| | DYS (n=20) | | CON (n=27) | | Test | p | Effect size | BF$_{10}$ |
|---|---|---|---|---|---|---|---|---|
| | **M** | **SD** | **M** | **SD** | **Test** | **p** | **Effect size** | **BF$_{10}$** |
| *Demographics* | | | | | | | | |
| Sex | 12 F, 8 M | | 11 F, 16 M | | $\chi$=1.71 | 0.192 | phi = −0.19 | 0.81 |
| Age | 19.98 | 3.92 | 20.33 | 3.25 | U=261.0 | 0.846 | $r_{rb}$ = 0.03 | 0.29 |
| Mother's education (years) | 17.48 | 3.68 | 16.63 | 2.56 | U=242.5 | 0.551 | $r_{rb}$ = −0.10 | 0.36 |
| Father's education (years) | 16.74[a] | 3.46[a] | 16.41 | 3.49 | U=237.0 | 0.659 | $r_{rb}$ = −0.08 | 0.31 |
| IQ | 103.80 | 14.30 | 113.56 | 9.23 | t(30.41)=2.67 | **0.012** | d=0.81 | 6.65 |
| Nonverbal IQ (scaled score) | 10.45 | 3.14 | 12.37 | 2.17 | U=174.5 | **0.038** | $r_{rb}$ = 0.35 | 1.79 |
| ARHQ-PL | 52.90 | 10.78 | 24.26 | 6.71 | U=6.0 | **<0.001** | $r_{rb}$ = −0.98 | 1060.02 |
| *Reading and reading-related tasks* | | | | | | | | |
| Words/min | 110.80 | 19.59 | 136.11 | 12.80 | U=74.0 | **<0.001*** | $r_{rb}$ = 0.73 | 108.53 |
| Pseudowords/min | 59.15 | 13.91 | 85.59 | 15.83 | t(45)=5.96 | **<0.001*** | d=1.76 | >10,000 |
| RAN objects (s) | 32.55 | 4.36 | 28.56 | 4.46 | U=122.0 | **0.001*** | $r_{rb}$ = −0.55 | 7.92 |
| RAN colors (s) | 36.40 | 4.91 | 30.63 | 3.67 | t(45)=−4.61 | **<0.001*** | d=−1.36 | 571.37 |
| RAN digits(s) | 19.35 | 4.00 | 16.33 | 2.24 | U=134.0 | **0.003*** | $r_{rb}$ = −0.50 | 3.82 |
| RAN letters (s) | 21.75 | 3.24 | 19.52 | 2.53 | t(45)=−2.65 | **0.011** | d=−0.78 | 4.52 |
| Reading comprehension (s) | 63.10 | 18.30 | 43.89 | 7.87 | U=82.0 | **<0.001*** | $r_{rb}$ = −0.70 | 65.88 |
| Phoneme deletion (% correct) | 74.62 | 29.20 | 94.16 | 6.61 | U=120.0 | **0.001*** | $r_{rb}$ = 0.56 | 13.46 |
| Spoonerisms phonemes (% correct) | 52.14 | 38.23 | 87.83 | 8.36 | U=106.5 | **<0.001*** | $r_{rb}$ = 0.61 | 17.43 |
| Spoonerisms syllables (% correct) | 41.67 | 29.86 | 77.78 | 20.67 | U=87.5 | **<0.001*** | $r_{rb}$ = 0.68 | 27.77 |
| Orthographic awareness (accuracy/time) | 0.35 | 0.16 | 0.54 | 0.13 | t(45)=4.26 | **<0.001*** | d=1.26 | 215.19 |
| Perception speed (sten score) | 2.95 | 1.79 | 4.33 | 1.64 | U=126.0 | **0.002*** | $r_{rb}$ = 0.53 | 4.49 |
| Digits forward | 5.65 | 1.79 | 7.04 | 2.03 | U=158.0 | **0.014** | $r_{rb}$ = 0.42 | 2.13 |
| Digits backward | 4.95 | 1.73 | 7.48 | 2.06 | t(45)=4.45 | **<0.001*** | d=1.31 | 355.64 |

Note: DYS – dyslexic group; CON – control group; F – females, M – males. BF$_{10}$ – Bayes factor indicating ratio of the likelihood of an alternative hypothesis (H1) to a null hypothesis (H0). ARHQ-PL – Polish version of the Adult Reading History Questionnaire. RAN – rapid automatized naming. Non-parametric Mann-Whitney test was performed when assumption of normal distribution was violated. $r_{rb}$ – rank biserial correlation provided as an effect size parameter for Mann-Whitney test. Boldface indicates statistical significance at p<0.05 level (uncorrected).

*Significance after Bonferroni correction for 14 planned comparisons for reading and reading-related tasks;

[a]n = 19 (one participant did not provide information about the father's education).

**Appendix 1—table 3.** Zero-order correlations between magnetic resonance spectroscopy (MRS) and electroencephalography (EEG) biomarkers of excitatory-inhibitory balance.

| Variable | 1. $r$ ($BF_{10}$) | 2. | 3. | 4. | 5. | 6. | 7. | 8. |
|---|---|---|---|---|---|---|---|---|
| *EEG resting state* | | | | | | | | |
| 1. Glu | – | | | | | | | |
| 2. GABA+ | 0.40**[a] (8.08) | – | | | | | | |
| 3. Glu/GABA +ratio | –0.11 (0.24)[a] | –0.90***[a] (>10,000) | – | | | | | |
| 4. Glu/GABA +imbalance | 0.16 (0.32)[a] | 0.33*[a] (2.14) | –0.19 (0.42)[a] | – | | | | |
| 5. Exponent mean (rest) | 0.13 (0.25)[b] | 0.10 (0.23)[a] | –0.11 (0.24)[a] | 0.31* (1.58)[a] | – | | | |
| 6. Offset mean (rest) | 0.10 (0.23)[b] | 0.17 (0.35)[a] | –0.15 (0.30)[a] | 0.26 (0.79)[a] | 0.70*** (>10,000)[c] | – | | |
| 7. Exponent left STS (rest) | 0.04 (0.18)[b] | 0.09 (0.22)[a] | –0.08 (0.21)[a] | 0.37* (4.06)[a] | 0.70*** (>10,000)[c] | 0.52*** (>10,000)[c] | – | |
| 8. Offset left STS (rest) | –0.09 (0.21)[b] | 0.01 (0.18)[a] | 0.00 (0.18)[a] | 0.21 (0.48)[a] | 0.24** (3.92)[c] | 0.54*** (>10,000)[c] | 0.70*** (>10,000)[c] | – |
| 9. Beta power left STS (rest) | –0.06 (0.19)[b] | 0.22 (0.53)[a] | –0.28 (1.04)[a] | 0.03 (0.19)[a] | 0.19* (0.85)[c] | 0.20* (1.29)[c] | 0.50*** (>10,000)[c] | 0.56*** (>10,000)[c] |
| *EEG language task* | | | | | | | | |
| 5. Exponent mean (task) | 0.14 (0.27)[b] | 0.15 (0.29)[a] | –0.14 (0.28)[a] | 0.24 (0.64)[a] | – | | | |
| 6. Offset mean (task) | 0.10 (0.23)[b] | 0.19 (0.41)[a] | –0.16 (0.31)[a] | 0.23 (0.59)[a] | 0.75*** (>10,000)[c] | – | | |
| 7. Exponent left STS (task) | 0.05 (0.19)[b] | 0.09 (0.22)[a] | –0.09 (0.21)[a] | 0.24 (0.64)[a] | 0.69*** (>10000)[c] | 0.58*** (>10,000)[c] | – | |
| 8. Offset left STS (task) | –0.07 (0.20)[b] | 0.03 (0.19)[a] | –0.01 (0.18)[a] | 0.18 (0.38)[a] | 0.36*** (375.45)[c] | 0.62*** (>10,000)[c] | 0.80*** (>10,000)[c] | – |
| 9. Beta power left STS (task) | –0.07 (0.20)[b] | 0.22 (0.52)[a] | –0.25 (0.77)[a] | –0.01 (0.18)[a] | 0.07 (0.15)[c] | 0.15 (0.43)[c] | 0.47*** (>10,000)[c] | 0.59*** (>10,000)[c] |

Note: $r$ – Pearson's correlation coefficient; $BF_{10}$ – Bayes factor indicating ratio of the likelihood of an alternative hypothesis (H1) to a null hypothesis (H0); mean – values averaged across all electrodes; left STS – values averaged across three electrodes corresponding to the left superior temporal sulcus (T7, TP7, TP9).

***p < 0.001 (uncorrected); **p < 0.01 (uncorrected); *p < 0.05 (uncorrected).

[a]$n = 47$; [b]$n = 50$; [c]$n = 119$.

**Appendix 1—table 4.** Zero-order correlations between reading, phonological awareness, rapid automatized naming, multisensory integration, and biomarkers of excitatory-inhibitory balance.

| Variable | 1. r (BF$_{10}$) | 2. | 3. | 4. |
|---|---|---|---|---|
| *EEG resting state* | | | | |
| 1. Reading | – | | | |
| 2. Phonological awareness | 0.62*** (>10,000)[c] | – | | |
| 3. RAN | 0.73*** (>10,000)[c] | 0.50*** (>10,000)[c] | – | |
| 4. Multisensory integration | 0.24* (1.44)[d] | 0.33** (16.94)[d] | 0.08 (0.17)[d] | – |
| 5. Glu | −0.10 (0.22)[b] | −0.16 (0.32)[b] | −0.08 (0.20)[b] | −0.08 (0.21)[b] |
| 6. GABA+ | −0.17 (0.35)[a] | −0.02 (0.18)[a] | −0.19 (0.41)[a] | 0.24 (0.62)[a] |
| 7. Glu/GABA+ ratio | 0.03 (0.19)[a] | −0.02 (0.18)[a] | 0.08 (0.21)[a] | −0.31* (1.62)[a] |
| 8. Glu/GABA+ imbalance | −0.21 (0.47)[a] | −0.06 (0.20)[a] | −0.19 (0.40)[a] | 0.13 (0.27)[a] |
| 9. Exponent mean (rest) | −0.13 (0.30)[c] | 0.05 (0.13)[c] | −0.08 (0.17)[c] | −0.05 (0.15)[d] |
| 10. Offset mean (rest) | −0.03 (0.12)[c] | 0.04 (0.13)[c] | −0.02 (0.12)[c] | −0.01 (0.14)[d] |
| 11. Exponent left STS (rest) | −0.14 (0.37)[c] | −0.01 (0.12)[c] | −0.07 (0.16)[c] | −0.15 (0.33)[d] |
| 12. Offset left STS (rest) | 0.03 (0.12)[c] | 0.09 (0.18)[c] | 0.03 (0.12)[c] | −0.07 (0.17)[d] |
| 13. Beta power left STS` (rest) | 0.03 (0.12)[c] | 0.22* (1.96)[c] | −0.04 (0.12)[c] | 0.04 (0.14)[d] |
| *EEG language task* | | | | |
| 9. Exponent mean (task) | −0.13 (0.32)[c] | 0.06 (0.14)[c] | −0.14 (0.34)[c] | −0.09 (0.19)[d] |
| 10. Offset mean (task) | −0.05 (0.13)[c] | 0.04 (0.12)[c] | −0.05 (0.13)[c] | −0.01 (0.13)[d] |
| 11. Exponent left STS (task) | −0.11 (0.23)[c] | 0.01 (0.12)[c] | −0.11 (0.24)[c] | −0.17 (0.48)[d] |
| 12. Offset left STS (task) | 0.04 (0.12)[c] | 0.09 (0.18)[c] | 0.01 (0.12)[c] | −0.07 (0.16)[d] |
| 13. Beta power left STS (task) | 0.05 (0.13)[c] | 0.21* (1.61)[c] | 0.02 (0.12)[c] | 0.11 (0.22)[d] |

Note: *r* – Pearson's correlation coefficient; *BF*$_{10}$ – Bayes factor indicating ratio of the likelihood of an alternative hypothesis (H1) to a null hypothesis (H0); mean – values averaged across all electrodes; left STS – values averaged across three electrodes corresponding to the left superior temporal sulcus (T7, TP7, TP9).

***p < 0.001 (uncorrected); **p < 0.01 (uncorrected); *p < 0.05 (uncorrected).

[a]*n* = 47; [b]n = 50; [c]n = 119; [d]n = 87.

**Appendix 1—table 5.** One-sample t-tests separately for visual (words>false fonts) and auditory runs (words>words backward) within control (CON) and dyslexic groups (DYS).

| Brain regions | Hemisphere | Peak of cluster coordinates | | | t-Value | Number of voxels |
|---|---|---|---|---|---|---|
| | | x | y | z | | |
| **CON (*n*=29) visual runs (FWEc = 127)** | | | | | | |
| Middle temporal gyrus, superior temporal gyrus, inferior frontal gyrus (pars triangularis, orbitalis, opercularis), supramarginal gyrus, temporal pole (superior and middle temporal gyri), postcentral gyrus, hippocampus, posterior orbital gyrus, amygdala, parahippocampal gyrus, inferior parietal gyrus, rolandic operculum, lateral orbital gyrus, pallidum, anterior orbital gyrus, inferior frontal gyrus, putamen, middle frontal gyrus, insula, angular gyrus | L | −62 | −54 | 10 | 9.78 | 5360 |
| Precuneus, posterior cingulate gyrus, middle cingulate and paracingulate gyri, calcarine fissure, cuneus | L/R | -4 | −52 | 20 | 7.17 | 957 |
| Middle temporal gyrus, superior temporal gyrus, temporal pole (superior and middle temporal gyri), inferior temporal gyrus | R | 60 | −34 | 0 | 7.10 | 1117 |
| Cuneus, superior occipital gyrus, calcarine fissure | L/R | 14 | −92 | 26 | 5.98 | 334 |
| Superior frontal gyrus (dorsolateral and medial) | L | -8 | 54 | 34 | 5.95 | 678 |
| Superior frontal gyrus (medial orbital), anterior cingulate cortex (pregenual), superior frontal gyrus (medial) | L/R | -8 | 50 | -8 | 5.33 | 194 |
| Insula, putamen, rolandic operculum, inferior frontal gyrus (pars opercularis) | L | −36 | 4 | 6 | 5.09 | 127 |
| Angular gyrus, middle temporal gyrus, middle occipital gyrus | L | −40 | −58 | 26 | 4.56 | 167 |
| **DYS (*n*=21) visual runs (FWEc = 665)** | | | | | | |
| Inferior frontal gyrus (pars triangularis, orbitalis, opercularis), posterior orbital gyrus | L | −48 | 24 | 0 | 6.54 | 665 |
| Middle temporal gyrus, superior temporal gyrus, supramarginal gyrus | L | −62 | −32 | 2 | 6.40 | 677 |
| **CON (*n*=29) auditory run (FWEc = 124)** | | | | | | |

*Appendix 1—table 5 Continued on next page*

*Appendix 1—table 5 Continued*

| Brain regions | Hemisphere | Peak of cluster coordinates | | | t-Value | Number of voxels |
|---|---|---|---|---|---|---|
| | | x | y | z | | |
| Middle temporal gyrus, postcentral gyrus, cingulate gyrus (mid part), superior parietal gyrus, precentral gyrus, precuneus, middle occipital gyrus, supramarginal gyrus, inferior parietal gyrus, inferior temporal gyrus, calcarine fissure, fusiform gyrus, superior temporal gyrus, angular gyrus, temporal pole (superior and temporal gyri), cuneus, supplementary motor area, lingual gyrus, inferior frontal gyrus (pars triangularis, orbitalis), posterior cingulate gyrus, superior frontal gyrus (dorsolateral), inferior occipital gyrus, posterior orbital gyrus, paracentral lobule, rolandic operculum, parahippocampal gyrus, lateral orbital gyrus, middle frontal gyrus, superior occipital gyrus, anterior orbital gyrus | L/R | –52 | –12 | –10 | 9.75 | 13,444 |
| Middle temporal gyrus, superior temporal gyrus, temporal pole (superior temporal gyrus), inferior temporal gyrus | R | 50 | –4 | –20 | 7.98 | 1038 |
| Putamen, insula, rolandic operculum, pallidum, Heschl's gyrus, amygdala, hippocampus, thalamus (lateral geniculate) | L | –34 | –20 | 8 | 6.15 | 463 |
| Thalamus (mediodorsal medial magnocellular, intralaminar, pulvinar medial, ventral posterolateral, mediodorsal lateral parvocellular, ventral lateral, pulvinar anterior, lateral posterior, pulvinar inferior, medial geniculate), hippocampus, lingual gyrus, parahippocampal gyrus | L | –10 | –20 | 0 | 5.82 | 259 |
| Superior frontal gyrus (dorsolateral and medial), middle frontal gyrus | L/R | –6, | 60 | 34 | 5.81 | 438 |

*Appendix 1—table 5 Continued on next page*

*Appendix 1—table 5 Continued*

| Brain regions | Hemisphere | Peak of cluster coordinates | | | t-Value | Number of voxels |
|---|---|---|---|---|---|---|
| | | x | y | z | | |
| Thalamus (ventral posterolateral, pulvinar medial, pulvinar anterior, lateral geniculate, intralaminar, pulvinar inferior, pulvinar lateral, ventral lateral, medial geniculate), hippocampus | R | 14 | −20 | 4 | 5.68 | 149 |
| Middle occipital gyrus, middle temporal gyrus, angular gyrus, superior occipital gyrus, superior temporal gyrus, superior parietal gyrus, cuneus, supramarginal gyrus, inferior parietal gyrus | R | 36 | −76 | 40 | 5.46 | 978 |
| Cerebellar hemispheres (lobules IV, V, VI, VIII, IX, crus I, crus II), vermis (lobules VI, VII, VIII, IX) | L/R | 12, | −50 | −30 | 5.35 | 630 |
| Inferior frontal gyrus (pars triangularis) | L | −50 | 22 | 20 | 4.54 | 124 |
| **DYS (*n*=21) auditory run** (FWEc = 192) | | | | | | |
| Middle temporal gyrus, superior temporal gyrus, temporal pole (superior temporal gyrus) | L | −52 | −12 | −12 | 7.00 | 403 |
| Middle temporal gyrus | L | −56 | −38 | 0 | 5.33 | 192 |

Note: L – left, R – right. All results are reported at p<0.001 height threshold corrected for multiple comparisons using p<0.05 FWE cluster threshold.

**Appendix 1—table 6.** Logical conjunction results from paired t-tests for both visual (words>false fonts) and auditory runs (words>words backward) within control (CON) and dyslexic groups (DYS).

| Brain regions | Hemisphere | Peak of cluster coordinates | | | t-Value | Number of voxels |
|---|---|---|---|---|---|---|
| | | x | y | z | | |
| **CON (*n*=29) visual and auditory runs conjunction** (FWEc = 143) | | | | | | |
| Middle temporal gyrus, inferior frontal gyrus (pars triangularis, orbitalis, opercularis), superior temporal gyrus, temporal pole (superior and middle temporal gyri), supramarginal gyrus, posterior orbital gyrus, inferior parietal gyrus, angular gyrus, lateral orbital gyrus, anterior orbital gyrus, middle occipital gyrus | L | −54 | −58 | 12 | 9.58 | 3164 |
| Middle temporal gyrus, superior temporal gyrus | R | 50 | −34 | −2 | 7.98 | 186 |

*Appendix 1—table 6 Continued on next page*

*Appendix 1—table 6 Continued*

| Brain regions | Hemisphere | Peak of cluster coordinates | | | t-Value | Number of voxels |
|---|---|---|---|---|---|---|
| | | x | y | z | | |
| Superior frontal gyrus (dorsolateral and medial) | L | -6 | 56 | 36 | 5.94 | 430 |
| Superior temporal gyrus, temporal pole (superior and middle temporal gyri), middle temporal gyrus | R | 52 | 10 | –16 | 5.76 | 175 |
| Precuneus, posterior cingulate gyrus, middle cingulate and paracingulate gyri | L/R | –2 | –56 | 30 | 5.48 | 448 |
| Supramarginal gyrus, postcentral gyrus, rolandic operculum, superior temporal gyrus | L | –50 | –24 | 22 | 5.35 | 217 |
| Middle cingulate and paracingulate gyri | L/R | –2 | –10 | 40 | 5.17 | 143 |
| **DYS (n=21) visual and auditory runs conjunction (FWEc = 545)** | | | | | | |
| Middle temporal gyrus, superior temporal gyrus | L | –52 | –12 | –6 | 6.73 | 545 |

Note: L – left, R – right. All results are reported at p<0.001 height threshold corrected for multiple comparisons using p<0.05 FWE cluster threshold.

**Appendix 1—table 7.** Results from the flexible factorial model for both visual (words>false fonts) and auditory runs (words>words backward) between control (CON) and dyslexic groups (DYS).

| Brain regions | Hemisphere | Peak of cluster coordinates | | | t-Value | Number of voxels |
|---|---|---|---|---|---|---|
| | | x | y | z | | |
| **CON>DYS main effect of group for visual and auditory runs (FWEc = 121)** | | | | | | |
| Middle temporal gyrus, supramarginal gyrus, superior temporal gyrus, inferior temporal gyrus, inferior parietal gyrus, middle occipital gyrus, fusiform gyrus, angular gyrus, postcentral gyrus, rolandic operculum | L | –58 | –30 | 30 | 7.19 | 2181 |
| Rolandic operculum, superior temporal gyrus, middle temporal gyrus, supramarginal gyrus, insula, postcentral gyrus, precentral gyrus, inferior frontal gyrus (pars opercularis), putamen, inferior temporal gyrus | R | 46 | –4 | 14 | 7.07 | 1512 |

*Appendix 1—table 7 Continued on next page*

*Appendix 1—table 7 Continued*

| Brain regions | Hemisphere | Peak of cluster coordinates | | | t-Value | Number of voxels |
|---|---|---|---|---|---|---|
| | | x | y | z | | |
| Superior frontal gyrus (dorsolateral and medial), middle frontal gyrus | R | 22 | 34 | 54 | 6.56 | 464 |
| Insula, rolandic operculum, superior temporal gyrus, Heschl's gyrus | L | –40 | –6 | 18 | 6.21 | 266 |
| Middle cingulate and paracingulate gyri, paracentral lobule, supplementary motor area, precuneus | L/R | –8 | –26 | 42 | 5.96 | 1108 |
| Precentral gyrus, superior frontal gyrus (dorsolateral) | L | –32 | –8 | 68 | 5.70 | 191 |
| Angular gyrus, supramarginal gyrus, inferior parietal gyrus, middle occipital gyrus | R | 52 | –62 | 40 | 5.37 | 516 |
| Superior frontal gyrus (dorsolateral and medial), middle frontal gyrus | L | –18 | 38 | 34 | 5.13 | 241 |
| Precentral gyrus, postcentral gyrus, middle frontal gyrus | R | 58 | –12 | 46 | 5.09 | 140 |
| Lingual gyrus, cerebellar hemisphere (lobule VI), fusiform gyrus | R | 14 | –68 | –10 | 5.04 | 209 |
| Temporal pole (superior temporal gyrus), middle temporal gyrus, superior temporal gyrus | L | –54 | 10 | –16 | 5.00 | 156 |
| Supplementary motor area, middle cingulate and paracingulate gyri | L/R | 4 | 6 | 48 | 4.93 | 174 |
| Postcentral gyrus, superior parietal gyrus | R | 24 | –48 | 58 | 4.85 | 121 |
| Postcentral gyrus, supramarginal gyrus | R | 38 | –32 | 36 | 4.78 | 137 |
| **DYS>CON main effect of group for visual and auditory runs** | | | | | | |
| No suprathreshold clusters | | | | | | |

Note: L – left, R – right. All results are reported at p<0.001 height threshold corrected for multiple comparisons using p<0.05 FWE cluster threshold.

**Appendix 1—table 8.** Magnetic resonance spectroscopy (MRS) checklist.

| **Hardware** | |
| --- | --- |
| Field strength [T] | 7T |
| Manufacturer | GE Healthcare |
| Model (software version if available) | Discovery MR 950 |
| RF coils: nuclei (transmit/receive), number of channels, type, body part | H$^1$ 32 channel head coil |
| Additional hardware | MR Safe Response PAD |

| **Acquisition** | |
| --- | --- |
| Pulse sequence | Semi-Laser |
| Volume of interest (VOI) locations | Left superior temporal sulcus (STS) |
| Nominal VOI size [cm$^3$, mm$^3$] | 15×15×15 mm$^3$ |
| Repetition time (TR), echo time (TE) [ms, s] | TR = 4000 ms TE = 28 ms |
| Total number of excitations or acquisitions per spectrum | 320 averages |
| Additional sequence parameters (spectral width in Hz, number of spectral points, frequency offsets) | 5000 Hz, 4096 points |
| Water suppression method | VAPOR |
| Shimming method, reference peak, and thresholds for 'acceptance of shim' chosen | Automated linear shims adjustment, 0 and first shim order only, water peak, <20 Hz |
| Triggering or motion correction method (respiratory, peripheral, cardiac triggering, incl. device used and delays) | N/A |

| **Data analysis methods and outputs** | |
| --- | --- |
| Analysis software | fsl-mrs (version 2.0.7) |
| Processing steps deviating from quoted reference or product | fsl_mrs default pipeline+simulated basis set |
| Output measure (e.g. absolute concentration, institutional units, ratio) Processing steps deviating from quoted reference or product | Ratio to total creatine |
| Quantification references and assumptions, fitting model assumptions | The customized basis set includes: Ala, Asc, Asp, Cit, Cr, EtOH, GABA, GPC, GSH, Glc, Gln, Glu, Gly, Ins, Lac, NAA, NAAG, PCh, PCr, PE, Phenyl, Scyllo, Ser, Tau, Tyros, bHB, bHG. Macromolecules MM09, MM12, MM14, MM17, MM21 were added. |

## Data quality

| Reported variables (SNR, linewidth (with reference peaks)) | *Linewidth (for metabolite group reported by fsl_mrs): mean 11.11 Hz, SD = 2.79 Hz*<br>*SNR (for NAA reported by fsl_mrs): mean 7.05, SD = 23.95* |
|---|---|
| Data exclusion criteria | *Linewidth >20 Hz, CRLB >20% and Visual inspection (baseline, residuals)*<br>*Glu – 4 subjects excluded GABA – 7 subjects excluded* |
| Quality measures of postprocessing Model fitting (e.g. CRLB, goodness of fit, SD of residual) | *%CRLB of Glu: mean 2.96, SD = 0.79*<br>*%CRLB of GABA: mean 10.59, SD = 2.76*<br>*%CRLB of NAA: 1.76 SD = 0.46* |

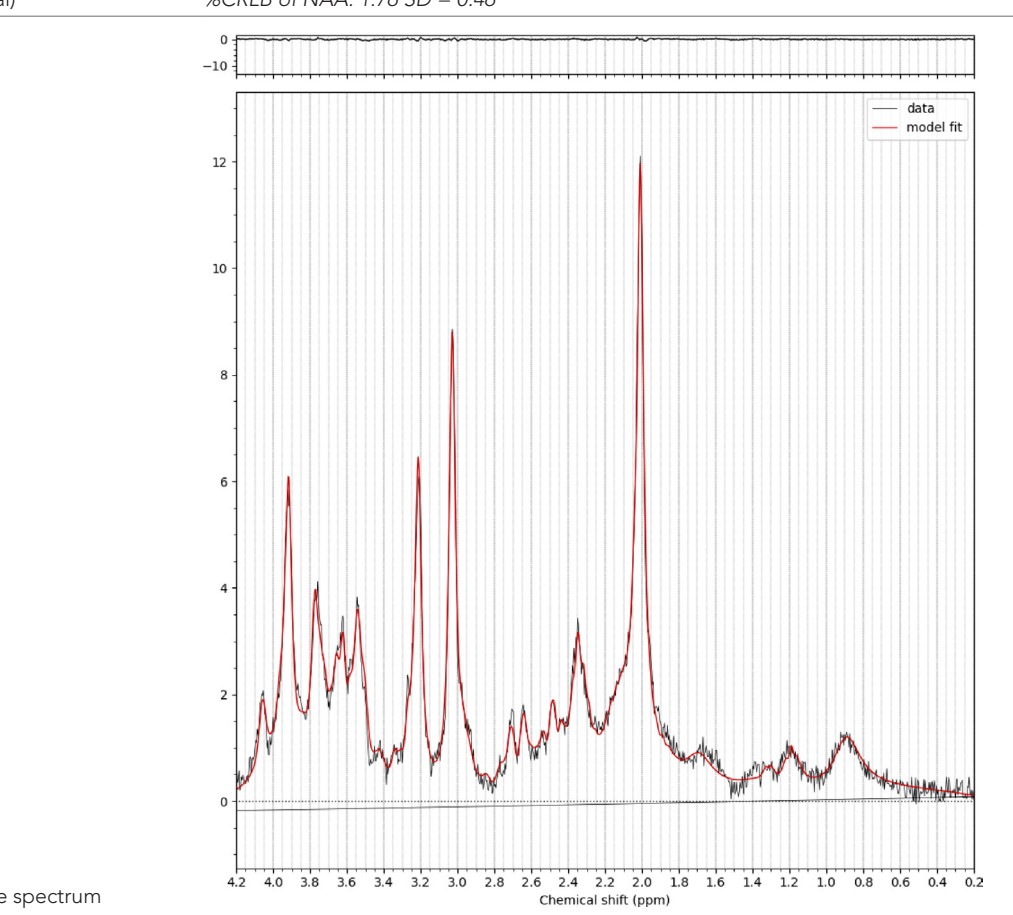

Sample spectrum

