## [Editor Report · eLife Assessment]

This study empirically investigates the neural noise hypothesis of developmental dyslexia using electroencephalography (EEG) during a spoken language task and 7T magnetic resonance spectroscopy (MRS). The **convincing** findings indicate no evidence of an imbalance between excitatory and inhibitory (E/I) brain activity in adolescents and young adults with dyslexia compared to controls, thereby challenging the neural noise hypothesis. This research is **valuable** for advancing our understanding of the neural mechanisms underlying dyslexia and offers broader insights into the neural processes involved in reading development.

---

## [Referee Report · Reviewer #1 (Public review)]

Summary:

"Neural noise", here operationalized as an imbalance between excitatory and inhibitory neural activity, has been posited as a core cause of developmental dyslexia, a prevalent learning disability that impacts reading accuracy and fluency. This is study is the first to systematically evaluate the neural noise hypothesis of dyslexia. Neural noise was measured using neurophysiological (electroencephalography [EEG]) and neurochemical (magnetic resonance spectroscopy [MRS]) in adolescents and young adults with and without dyslexia. The authors did not find evidence of elevated neural noise in the dyslexia group from EEG or MRS measures, and Bayes factors generally informed against including the grouping factor in the models. Although the comparisons between groups with and without dyslexia did not support the neural noise hypothesis, a mediation model that quantified phonological processing and reading abilities continuously revealed that EEG beta power in the left superior temporal sulcus was positively associated with reading ability via phonological awareness. This finding lends support for analysis of associations between neural excitatory/inhibitory factors and reading ability along a continuum, rather than as with a case/control approach, and indicates the relevance of phonological awareness as an intermediate trait that may provide a more proximal link between neurobiology and reading ability. Further research is needed across developmental stages and over a broader set of brain regions to more comprehensively assess the neural noise hypothesis of dyslexia, and alternative neurobiological mechanisms of this disorder should be explored.

Strengths:

The inclusion of multiple methods of assessing neural noise (neurophysiological and neurochemical) is a major advantage of this paper. MRS at 7T confers an advantage of more accurately distinguishing and quantifying glutamate, which is a primary target of this study. In addition, the subject-specific functional localization of the MRS acquisition is an innovative approach. MRS acquisition and processing details are noted in the supplementary materials using according to the experts' consensus recommended checklist (https://doi.org/10.1002/nbm.4484). Commenting on rigor the EEG methods is beyond my expertise as a reviewer.

Participants recruited for this study included those with a clinical diagnosis of dyslexia, which strengthens confidence in the accuracy of the diagnosis. The assessment of reading and language abilities during the study further confirms the persistently poorer performance of the dyslexia group compared to the control group.

The correlational analysis and mediation analysis provide complementary information to the main case-control analyses, and the examination of associations between EEG and MRS measures of neural noise is novel and interesting.

The authors follow good practice for open science, including data and code sharing. They also apply statistical rigor, using Bayes Factors to support conclusions of null evidence rather than relying only on non-significant findings. In the discussion, they acknowledge the limitations and generalizability of the evidence and provide directions for future research on this topic.

Appraisal:

The authors present a thorough evaluation of the neural noise hypothesis of developmental dyslexia in a sample of adolescents and young adults using multiple methods of measuring excitatory/inhibitory imbalances as an indicator of neural noise. The authors concluded that there was not support for the neural noise hypothesis of dyslexia in their study based on null significance and Bayes factors. This conclusion is justified, and further research is called for to more broadly evaluate the neural noise hypothesis in developmental dyslexia.

Impact:

This study provides an exemplar foundation for the evaluation of the neural noise hypothesis of dyslexia. Other researcher may adopt the model applied in this paper to examine neural noise in various populations with/without dyslexia, or across a continuum of reading abilities, to more thoroughly examine evidence (or lack thereof) for this hypothesis. Notably, the lack of evidence here does not rule out the possibility for a role of neural noise in dyslexia, and the authors point out that presentation with co-occurring conditions, such as ADHD, may contribute to neural noise in dyslexia. Dyslexia remains a multi-faceted and heterogenous neurodevelopmental condition, and many genetic, neurobiological and environmental factors play a role. This study demonstrates one step toward evaluating neurobiological mechanisms that may contribute to reading difficulties.

---

## [Referee Report · Reviewer #2 (Public review)]

Summary:

This study utilized two complimentary techniques (EEG and 7T MRI/MRS) to directly test a theory of dyslexia: the neural noise hypothesis. The authors report finding no evidence to support an excitatory/inhibitory balance, as quantified by beta in EEG and Glutamate/GABA ratio in MRS. This is important work and speaks to one potential mechanism by which increased neural noise may occur in dyslexia.

Strengths:

This is a well-conceived study with in depth analyses and publicly available data for independent review. The authors provide transparency with their statistics and display the raw data points along with the averages in figures for review and interpretation. The data suggest that an E/I balance issue may not underlie deficits in dyslexia and is a meaningful and needed test of a possible mechanism for increased neural noise.

Weaknesses:

The researchers did not include a visual print task in the EEG task, which limits analysis of reading specific regions such as the visual word form area, which is a commonly hypoactivated region in dyslexia. This region is a common one of interest in dyslexia, yet the researchers measured the I/E balance in only one region of interest, specific to the language network.

---

## [Referee Report · Reviewer #3 (Public review)]

Summary:

This study by Glica and colleagues utilized EEG (i.e., Beta power, Gamma power, and aperiodic activity) and 7T MRS (i.e., MRS IE ratio, IE balance) to reevaluate the neural noise hypothesis in Dyslexia. Supported by Bayesian statistics, their results show convincing evidence of no differences in EI balance between groups, challenging the neural noise hypothesis.

Strengths:

Combining EEG and 7T MRS, this study utilized both the indirect (i.e., Beta power, Gamma power, and aperiodic activity) and direct (i.e., MRS IE ratio, IE balance) measures to reevaluate the neural noise hypothesis in Dyslexia.

---

## [Author Response]

The following is the authors’ response to the previous reviews.

**Reviewer #1 (Public review):**
Summary:"Neural noise", here operationalized as an imbalance between excitatory and inhibitory neural activity, has been posited as a core cause of developmental dyslexia, a prevalent learning disability that impacts reading accuracy and fluency. This is study is the first to systematically evaluate the neural noise hypothesis of dyslexia. Neural noise was measured using neurophysiological (electroencephalography [EEG]) and neurochemical (magnetic resonance spectroscopy [MRS]) in adolescents and young adults with and without dyslexia. The authors did not find evidence of elevated neural noise in the dyslexia group from EEG or MRS measures, and Bayes factors generally informed against including the grouping factor in the models. Although the comparisons between groups with and without dyslexia did not support the neural noise hypothesis, a mediation model that quantified phonological processing and reading abilities continuously revealed that EEG beta power in the left superior temporal sulcus was positively associated with reading ability via phonological awareness. This finding lends support for analysis of associations between neural excitatory/inhibitory factors and reading ability along a continuum, rather than as with a case/control approach, and indicates the relevance of phonological awareness as an intermediate trait that may provide a more proximal link between neurobiology and reading ability. Further research is needed across developmental stages and over a broader set of brain regions to more comprehensively assess the neural noise hypothesis of dyslexia, and alternative neurobiological mechanisms of this disorder should be explored.Strengths:The inclusion of multiple methods of assessing neural noise (neurophysiological and neurochemical) is a major advantage of this paper. MRS at 7T confers an advantage of more accurately distinguishing and quantifying glutamate, which is a primary target of this study. In addition, the subject-specific functional localization of the MRS acquisition is an innovative approach. MRS acquisition and processing details are noted in the supplementary materials using according to the experts' consensus recommended checklist (https://doi.org/10.1002/nbm.4484). Commenting on rigor the EEG methods is beyond my expertise as a reviewer.Participants recruited for this study included those with a clinical diagnosis of dyslexia, which strengthens confidence in the accuracy of the diagnosis. The assessment of reading and language abilities during the study further confirms the persistently poorer performance of the dyslexia group compared to the control group.The correlational analysis and mediation analysis provide complementary information to the main case-control analyses, and the examination of associations between EEG and MRS measures of neural noise is novel and interesting.The authors follow good practice for open science, including data and code sharing. They also apply statistical rigor, using Bayes Factors to support conclusions of null evidence rather than relying only on non-significant findings. In the discussion, they acknowledge the limitations and generalizability of the evidence and provide directions for future research on this topic.Weaknesses:Though the methods employed in the paper are generally strong, the MRS acquisition was not optimized to quantify GABA, so the findings (or lack thereof) should be interpreted with caution. Specifically, while 7T MRS affords the benefit of quantifying metabolites, such as GABA, without spectral editing, this quantification is best achieved with echo times (TE) of 68 or 80 ms in order to minimize the spectral overlap between glutamate and GABA and reduce contamination from the macromolecular signal (Finkelman et al., 2022, https://doi.org/10.1016/j.neuroimage.2021.118810). The data in the present study were acquired at TE=28 ms, and are therefore likely affected by overlapping Glu and GABA peaks at 2.3 ppm that are much more difficult to resolve at this short TE, which could directly affect the measures that are meant to characterize the Glu/GABA+ ratio/imbalance. In future research, MRS acquisition schemes should be optimized for the acquisition of Glutamate, GABA, and their relative balance.As the authors note in the discussion, additional factors such as MRS voxel location, participant age, and participant sex could influence associations between neural noise and reading abilities and should be considered in future studies.

We have modified Figure 2 and revised the paragraph discussing the MRS methodological limitations in accordance with Reviewer #1's recommendations. Additionally, we have included the CRLB and linewidth thresholds in the Results section. Furthermore, a new figure showing the correlations between EEG and MRS biomarkers has been added (Figure 3).

Appraisal:The authors present a thorough evaluation of the neural noise hypothesis of developmental dyslexia in a sample of adolescents and young adults using multiple methods of measuring excitatory/inhibitory imbalances as an indicator of neural noise. The authors concluded that there was not support for the neural noise hypothesis of dyslexia in their study based on null significance and Bayes factors. This conclusion is justified, and further research is called for to more broadly evaluate the neural noise hypothesis in developmental dyslexia.Impact:This study provides an exemplar foundation for the evaluation of the neural noise hypothesis of dyslexia. Other researcher may adopt the model applied in this paper to examine neural noise in various populations with/without dyslexia, or across a continuum of reading abilities, to more thoroughly examine evidence (or lack thereof) for this hypothesis. Notably, the lack of evidence here does not rule out the possibility for a role of neural noise in dyslexia, and the authors point out that presentation with co-occurring conditions, such as ADHD, may contribute to neural noise in dyslexia. Dyslexia remains a multi-faceted and heterogenous neurodevelopmental condition, and many genetic, neurobiological and environmental factors play a role. This study demonstrates one step toward evaluating neurobiological mechanisms that may contribute to reading difficulties.
**Reviewer #2 (Public review):**
Summary:This study utilized two complimentary techniques (EEG and 7T MRI/MRS) to directly test a theory of dyslexia: the neural noise hypothesis. The authors report finding no evidence to support an excitatory/inhibitory balance, as quantified by beta in EEG and Glutamate/GABA ratio in MRS. This is important work and speaks to one potential mechanism by which increased neural noise may occur in dyslexia.Strengths:This is a well conceived study with in depth analyses and publicly available data for independent review. The authors provide transparency with their statistics and display the raw data points along with the averages in figures for review and interpretation. The data suggest that an E/I balance issue may not underlie deficits in dyslexia and is a meaningful and needed test of a possible mechanism for increased neural noise.Weaknesses:The researchers did not include a visual print task in the EEG task, which limits analysis of reading specific regions such as the visual word form area, which is a commonly hypoactivated region in dyslexia. This region is a common one of interest in dyslexia, yet the researchers measured the I/E balance in only one region of interest, specific to the language network.
**Reviewer #3 (Public review):**
Summary:This study by Glica and colleagues utilized EEG (i.e., Beta power, Gamma power, and aperiodic activity) and 7T MRS (i.e., MRS IE ratio, IE balance) to reevaluating the neural noise hypothesis in Dyslexia. Supported by Bayesian statistics, their results show convincing evidence of no differences in EI balance between groups, challenging the neural noise hypothesis.Strengths:Combining EEG and 7T MRS, this study utilized both the indirect (i.e., Beta power, Gamma power, and aperiodic activity) and direct (i.e., MRS IE ratio, IE balance) measures to reevaluating the neural noise hypothesis in Dyslexia.